# Genes Containing Long Introns Occupy Series of Bands and Interbands in *Drosophila melanogaster* Polytene Chromosomes

**DOI:** 10.3390/genes11040417

**Published:** 2020-04-11

**Authors:** Varvara A. Khoroshko, Galina V. Pokholkova, Victor G. Levitsky, Tatyana Yu. Zykova, Oksana V. Antonenko, Elena S. Belyaeva, Igor F. Zhimulev

**Affiliations:** 1Department of the Chromosome Structure and Function, Laboratory of Molecular Cytogenetics, Institute of Molecular and Cellular Biology SB RAS, 630090 Novosibirsk, Russia; galina@mcb.nsc.ru (G.V.P.); vatolina@mcb.nsc.ru (T.Y.Z.); ovant@mcb.nsc.ru (O.V.A.); belyaeva@mcb.nsc.ru (E.S.B.); zhimulev@mcb.nsc.ru (I.F.Z.); 2Department of Systems Biology, Laboratory of Evolutionary Bioinformatics and Theoretical Genetics, Institute of Cytology and Genetics SB RAS, 630090 Novosibirsk, Russia; levitsky@bionet.nsc.ru; 3Department of Natural Sciences, Novosibirsk State University, 30090 Novosibirsk, Russia

**Keywords:** *Drosophila melanogaster*, polytene chromosomes, bands, interbands, housekeeping genes, chromatin states, gene introns

## Abstract

The *Drosophila melanogaster* polytene chromosomes are the best model for studying the genome organization during interphase. Despite of the long-term studies available on genetic organization of polytene chromosome bands and interbands, little is known regarding long gene location on chromosomes. To analyze it, we used bioinformatic approaches and characterized genome-wide distribution of introns in gene bodies and in different chromatin states, and using fluorescent in situ hybridization we juxtaposed them with the chromosome structures. Short introns up to 2 kb in length are located in the bodies of housekeeping genes (grey bands or *lazurite* chromatin). In the group of 70 longest genes in the *Drosophila* genome, 95% of total gene length accrues to introns. The mapping of the 15 long genes showed that they could occupy extended sections of polytene chromosomes containing band and interband series, with promoters located in the interband fragments (*aquamarine* chromatin). Introns (*malachite* and *ruby* chromatin) in polytene chromosomes form independent bands, which can contain either both introns and exons or intron material only. Thus, a novel type of the gene arrangement in polytene chromosomes was discovered; peculiarities of such genetic organization are discussed.

## 1. Introduction

Studies of the band and interband genetic organization simultaneously began with the discovery of polytene chromosomes in *Drosophila* in the mid-1930s, and the main hypotheses were soon formulated. For a long time, the detailed structure of genes, their exact chromosome localization, and functional role (developmental or housekeeping) remained unknown.

According to the hypothesis of N. Koltzoff [1], genes are located in the interbands, and bands are inactive chromosome structures, in which crossing over occurs. O. Mackensen [2], H. Muller, and A. Prokofyeva [3] localized the *white*, *vermilion*, and *scute* genes in the chromosome bands, therefore the bands were considered to be gene carriers. Some of the researchers believed that one gene corresponds to one band [2,4]; some supposed that bands could contain different number of genes, which in *Drosophila* genome could be estimated from five thousand to ten thousand and this roughly complies with the number of bands [5,6]. C. Metz believed that the gene could be partly located in the band and partly in the interband [7].

In 1950–1960, genes were localized on chromosomes in the regions using chromosome rearrangements method, the number of which was estimated to be approximately equal to the number of bands [8]. Many researchers assumed that the presence of a single gene in a band can be confirmed by the morphological manifestation of the gene activation in the form of a puff [9,10,11,12,13]. This hypothesis led W. Beermann to conclude that a band on the polytene chromosome is the physical equivalent of the Mendelian gene [14,15]. However, in the mid-1960s, G. Rudkin estimated the average band size to be 30–40 kb, which stipulated that, in one band, there is enough DNA for thirty 1 kb long genes, thus “one band–one gene” rule is untenable [16]. In connection with these conclusions, hypotheses based on the polygenicity of chromomeres began to appear. Yu. Olenov suggested that groups of genes active in the same tissue are located in bands to explain the obvious redundancy of DNA [17]. At the same time, B. Judd saturated a small region containing thinnest grey bands with mutations; following complementation analysis led the researches to the conclusion that numerical genes and bands ratio in the region is 1:1 [18] (for details see [19]).

In all of the described above gene models interbands played no role; according to the authors, the bands represented independent functional units specifically organized for a single gene or a gene group activity, and the appearance of models that considered the band and the adjoining interband as a single functional unit was unusual. F. Crick [20] believed that most of the DNA coding regions are located in the interbands; according to this hypothesis, the gene regulatory sites are located in the bands. On the contrary, J. Paul [21] and V. Sorsa [22] assumed that the interbands contain the main binding sites for RNA polymerase and, thus, the transcription starts in the interband initiation sites and is directed into bands from there. After the housekeeping genes concept was stated, some researches hypothesized the existence of genes that are active in all cells and they are constantly involved in transcription. Such genes are located in decondensed interbands and developmental genes are located in bands, since the pattern of bands is mostly constant [23,24].

Beginning with C. Bridges, three types of structures were described in polytene chromosomes: interbands and two types of bands—large densely packed black bands and loose light ones that look grey under the light microscope [25,26,27]. In addition to morphology, the two types of bands differ in the degree of chromatin compaction level, replication time, gene density, the presence or absence of replication origins, the sets of proteins, histone modifications, and chromatin states (see below for details).

After the *Drosophila* genome sequencing was completed [28] and the modENCODE program (model organism ENCyclopedia Of DNA Elements) identified genome elements [29], various proteins, nucleosomes, replication origins, histones, and their modifications were localized, according to which several chromatin classifications were presented [30,31,32] (for more details, see [33,34]).

The development of molecular genetic methods of labeling P-element interband insertions helped to determine which types of proteins related to bands and interbands in general [27,35,36,37]. The 4HMM mathematical model was developed using these data; it revealed four chromatin states that are distinguished by the presence of RNA polymerase II, Origin Recognition Complex (ORC), protein composition, and nucleosome modifications [38,39,40]. *Aquamarine* is predominantly localized in the polytene chromosome interbands, contains promoters of housekeeping genes and it is characterized by the specific set of proteins. This state is similar to “YELLOW” and “RED” states in the study of Filion et al. [30] and state “1” (TSS-proximal regions) in the study of Kharchenko et al. [31]. Gene exons along with several proteins and histones characteristic of transcription elongation are located within the *lazurite* fragments [33,38,41]. This state is similar to “YELLOW” state in the study of Filion et al. [30] and state “2” (transcriptional elongation marks) in the study of Kharchenko et al. [31]. *Aquamarine* and *lazurite* are both active states, which quite accurately correspond to the areas of open chromatin with condensation level “+1” in the study of Milon et al. [32]. *Malachite* always borders ruby fragments on both band edges and seems to be the point of transition between tightly compacted band material and decompacted active interband chromatin; this chromatin state was shown to be intermediate in terms of replication timing [42]. This state is similar to “BLACK” and “BLUE” states in the study of Filion et al. [30], state “9” (extensive silent domains) in the study of Kharchenko et al. [31], and it corresponds to neutral chromatin with condensation level “0” in the study of Milon et al. [32]. *Ruby* contains developmental genes and a specific set of proteins characteristic for inactivated chromatin. This state is similar to “BLACK” state in the study of Filion et al. [30], states “8” and “9” (heterochromatin-like regions containing moderate levels of H3K9me2/me3 and extensive silent domains) in the study of Kharchenko et al. [31] and corresponds to closed chromatin with condensation level “-1” in the study of Milon et al. [32]. The boundaries of four chromatin states fragments with a certain level of accuracy can be considered to be the boundaries of bands and interbands since each of the chromatin states is quite accurately localized in certain chromosome structures.

Recently, it was showed that interbands in conjunction with two types of bands display the portrait of the functioning genome. The housekeeping genes occupy two chromosome structures: while promoters of these genes are located in the interbands, the adjacent grey bands hold the gene bodies. Developmental genes are located in the dense black bands, which are usually polygenic. It was demonstrated that, in the thinnest bands, DNA is enough predominantly for a single housekeeping gene, so the B. Judd’s hypothesis “one gene–one band” was partly confirmed [40].

The length of an average band in *Drosophila* polytene chromosomes is about 30 kb according to various estimates [16,43], and the smallest distinguishable band contains 5 kb of DNA [44], as was already mentioned. When compared with the DNA length in the bands, introns up to 400 kb are quite large and, therefore, must somehow be detected at the chromosome level; at least, it can be expected that structures that are based on such long introns can be seen under the light microscope, but such data are unavailable in literature. According to our recently obtained preliminary data, certain large genes can occupy extended sections of chromosomes [45,46]. In the present study, we examined the localization sites of the genes that are mentioned above; in addition to that, we picked 12 new regions and analyzed their gene, intron, and chromatin composition, as well as studied their cytological location. We applied a simple approach, by using FISH, we mapped the start and the end of each long gene (usually the longest transcript). Subsequently, we studied the distribution of transcripts, introns, exons, chromatin states, and location of other genes, according to chromosome structures, such as interbands, black and grey bands.

## 2. Materials and Methods 

### 2.1. Fly Stock

In the present study, we used the fly stock with mutations in genes *white* and *SuUR^ES^* (*Suppressor of Under Replication*). The flies were kept on standard cornmeal-yeast-agar-molasses medium and larvae were grown under uncrowded conditions in standard test tubes at 18 °C for prolonged development [47].

### 2.2. Fluorescence In Situ Hybridization

Salivary glands of third instar larvae were dissected in Ephrussi–Beadle solution and transferred to a small glass cup with a mixture of ethanol and acetic acid (3:1). After 20 min fixation at room temperature, salivary glands were squashed in 45% acetic acid, frozen in liquid nitrogen, and stored in 70% ethanol at −20 °C. Fluorescence in situ hybridization (FISH) on polytene chromosomes was performed as described in [48]. Thirty-five DNA probes were obtained by standard PCR and then labeled with Flu-12-dUTP or Tamra-5-dUTP (Biosan) in random-primed polymerase reaction with the Klenow Fragment. Appendix A lists probes used in this study. Chromosome squashes were analyzed while using epifluorescence optics (Olympus BX50 microscope) and photographed with CCD Olympus DP50. For every probe colocalization, at least 50 nuclei on several slides were analyzed.

### 2.3. Bioinformatic Methods

In the present study we discuss only protein-coding genes; tRNA are grouped in clusters of 18S and 28S rRNA and, as a rule, cannot occupy a chromosome region larger than one band or a nuclear organizer [19]. The long non-coding RNA (lncRNA) genes [49] are not considered here because nothing is known regarding their location in polytene chromosome bands or interbands of *Drosophila*.

Using the full *D. melanogaster* genome annotation (Release 5.57), we identified transcription and translation start and termination sites for 26,938 transcripts with one or more introns; these data conforms to 11,632 genes. Based on these data, we pinpointed start and end positions for three types of introns, matching with transcripts 5’- and 3’-untranslated regions (UTR) and the remaining protein-coding sequences (gene bodies). The 5’UTR, gene bodies, and 3’UTR sampling groups consisted of 16782, 125,304, and 3125 introns accordingly. For the further analysis, we used the four chromatin-state model (4HMM), which comprises *aquamarine* (5748), *lazurite* (4139), *malachite* (9189), and *ruby* domains (6025) in the genome of *D. melanogaster* [38,39]. In order to estimate the number of genes when calculating the distribution of genes containing introns in 5’UTR, gene bodies, and 3’UTR by introns coverage with four chromatin states (Figure 1), we determined the equal weights *W* of introns taking into account the number of introns *NI*(*T*,*G*) of each transcript *T* of each *G* gene, as well as the number of transcripts of each *NT*(*G*) gene:*W* = 1/{*NI*(*T*,*G*) * *NT*(*G*)},(1)

For the four intron length ranges (<2 kb, 2–50 kb, 50–100 kb, >100 kb), we estimated the distribution of genes, containing introns in 5’UTR, gene bodies, 3’UTR, and overlapped them with four chromatin states (Figure 1). For each intron, the following formula was used:*Number of genes* = *K* * *W* * (*L_domain*/*L_total*),(2)
where *K* = 1 if the intron hits the length range (otherwise *K* = 0), *L*_*domain* equals intron length, which overlaps with the domain of given chromatin state, *L*_*total* equals the total intron length. We ranked the introns of 5’UTR, gene bodies, and 3’UTR by increasing lengths and determined the dependence of the introns fractions covered by four chromatin states on the intron length to determine the dependence of the introns fractions covered by four chromatin states on length (Figure 2). The calculations were carried out using moving average on the number of introns equaling 500.

## 3. Results

### 3.1. Characteristics of Introns and Genes Located In Different Chromatin Domains In The Drosophila Genome

Long introns can form bands in polytene chromosomes of *Drosophila melanogaster*; with this in mind, we studied the distribution and occurrence of genes containing introns, and characteristics of their localization in polytene chromosomes. While using FlyBase Release 5.57, we analyzed length distribution of 13,832 genes and their introns length in the *Drosophila* genome (Figure 1). With increase in gene length, total gene introns size also increases, as the result the longest genes are almost entirely composed of intron material (Figure 1A). We found that the intron DNA length significantly and positively correlates with the length of genes (Kendall’s rank correlation coefficient r = 0.494, p < 1E^−973^) (Figure 1A).

We divided the genes into four groups depending on the length of their largest transcript to relate the data to gene localization in different 4HMM model chromatin states: lengths up to 50 kb, 50–100 kb, 100–150 kb, and over 150 kb. The groups contained 13547, 214, 55, and 16 genes (Figure 1B). We found that, with increasing gene length, the coverage with *ruby* (known as repressed chromatin in which the developmental genes are located) increases from 30% for gene lengths up to 50 kb, to 55%, 57%, and 66% for gene lengths in the ranges 50–100 kb, 100–150 kb, and over 150 kb, accordingly (Figure 1B). At the same time, the *malachite* chromatin coverage changes insignificantly (22%, 26%, 22%, and 22%). The active states of chromatin corresponding to interbands and loose bands (promoters and exons of active genes) represent a total of 45% (17% and 28% for *aquamarine* and *lazurite*); while, for genes with a longer length, their proportion does not exceed 3% (Figure 1B). Therefore, the increase of genes length correlates with intron length increase, thus the proportion of repressed chromatin *ruby* coverage grows, and the proportion of active *aquamarine* and transcribed *lazurite* chromatin states reduces.

We analyzed introns within the regions between translation start and termination sites, 5′- and 3′-untranslated regions in genes located in four chromatin states (Figure 2). The most common gene bodies contain short introns of less than 2 kb in length, which overlap with *lazurite* chromatin (3380 genes); less often they overlap with *aquamarine* (2326 genes) and *malachite* (1505 gens). Fifty-six genes from this group contain introns, which do not overlap with any chromatin states (Figure 2A). Gene bodies also contain longer introns varying from 2 to 50 kb in length; however, such genes in the *Drosophila* genome are scarce (418). The introns of this group of genes mainly overlap with *ruby* (180 genes) and *malachite* chromatin (88 genes) (Figure 2A). We found only six genes with introns larger than 50 kb, and genes with introns larger than 100 kb are completely absent (FlyBase). In the *Drosophila* genome there are 1069 and 243 genes that contain introns of less than 2 kb in 5’UTR and 3’UTR, as well as 301 and 20 genes that contain introns from 2 to 50 kb in 5’UTR and 3’UTR, respectively (Figure 2B,C).

Each intron fragment contains several chromatin states and, with the introns length increase, the proportion of chromatin of various states changes (Figure 3). The size of introns in the gene body varies from 25 to 139,349 bp (mean 1225, median 82 bp) and it is mainly represented by short introns up to 2 kb in length, being predominantly enriched in *lazurite* chromatin. In introns larger than 2 kb, the *lazurite* fraction decreases and such introns are enriched in *ruby* and *malachite* chromatin. Introns that are larger than 5 kb also include a part of the genome with data being unavailable due to the 4HMM model chromatin states mosaic pattern (Figure 3A). The size of introns in 5’UTR on average vary from 43 to 133,766 bp (mean 4220, median 1120 bp) and are represented by introns up to 2 kb in length, mainly enriched in *aquamarine* chromatin. In introns larger than 2 kb, a decrease in the proportion of *aquamarine* and enrichment with *ruby* and *malachite* chromatin is observed, whereas *lazurite* chromatin is practically absent in 5’UTR of gene introns. Introns that are larger than 6 kb include a part of genome that lacks data for 4HMM chromatin states (Figure 3B). Introns in 3’UTR vary from 46 to 41,975 bp in length (mean 667, median 88 bp) and they are enriched in *ruby* chromatin (Figure 3C).

Figure 4 shows the overlap between four chromatin states and the intron coverage in gene bodies, in 5’UTR and 3’UTR. The total length of introns in gene bodies comprises 153.2 Mb; *aquamarine*, *lazurite*, *malachite*, and *ruby* chromatin cover 10.7 Mb, 23.5 Mb, 30.8 Mb, and 53 Mb respectively. A significant proportion of introns in the gene bodies (35.5 Mb) includes a part of the genome for which data on chromatin states is absent (Figure 4B). Total length of introns in 5’UTR comprises 70.8 Mb; *lazurite*, *malachite*, and *ruby* chromatin cover 2.9 Mb, 15.6 Mb, and 25.9 Mb respectively. For 16.9 Mb of introns in 5’UTR data on chromatin states is absent (Figure 4A). The total length of introns in 3’UTR comprises 2.1 Mb; *aquamarine*, *malachite*, *lazurite*, and *ruby* chromatin cover 0.1 Mb, 0.4 Mb, 0.5 Mb, and 0.8 Mb, respectively. For 0.3 Mb of introns in 3’UTR data on chromatin states is absent (Figure 4C). As a result, introns in the gene bodies, in 5’UTR and 3’UTR are mainly represented by *ruby* and *malachite* chromatin (developmental genes localization); in gene bodies, introns also overlap with *lazurite* chromatin. A rather significant proportion of introns corresponds to a part of the genome for which data on 4HMM chromatin states are absent.

In *Drosophila* genome, we found 70 genes with length varying from 100 up to 395 kb; the gene intron coverage stand over 95% for 31 genes, 93% for 52, 90% for 63, and 64% and 60% for two genes, respectively (Appendix A). Thus, the gene length increase relates to the enormous increase of intron length. Figure 5A,B shows the 70 genes overlap with four chromatin states (4HMM) and three chromatin condensation states (3CM). The analysis showed that genes are mostly composed of *ruby* (60.1%) and *malachite* chromatin (22.1%). The active chromatin states that correspond to interbands, promoters, and coding fragments of active genes total 6.5% (*aquamarine* 3.0% and *lazurite* 3.6% accordingly). For 11.7% of total gene length, the model does not provide data on chromatin states distribution (Figure 5A). The data on four chromatin states coverage correlate with the result on chromatin compaction level: closed and compact chromatin totals 62.4%, neutral 31.9%, and open active chromatin 5.5%, respectively (Figure 5B). Thus, we can conclude that all 70 studied genes are developmental and localized in black bands.

From the list of 70 genes, we picked three (Appendix A) and other 12 developmental genes, which had the longest length and the highest intron coverage, some of which are involved into ecdysone cascade regulation (Table 1). Figure 5C,D show the 15 genes overlap with four chromatin states (4HMM) and three chromatin condensation states (3CM). Introns correspond to *malachite* chromatin, which covers 41.9% and inactive state *ruby* covers 28.2% of total gene length. Active states corresponding to interbands, promoters and exons of active genes total 22.5% (*aquamarine* 12.1% and *lazurite* 10.3%, accordingly). For 7.2% of total gene length, the model does not provide data on chromatin states distribution (Figure 5C). Introns in 15 studied genes mostly overlap with neutral and closed chromatin condensation state and they correspond to 38.8% and 41.8% of total gene length accordingly; open chromatin state equals 19.3% (Figure 5D).

### 3.2. Long Genes Mapping In Polytene Chromosomes Bands and Interbands

#### 3.2.1. *CG3777*

*CG3777* is located on chromosome X in the 1A1–1A5 region (according to UCSC genome browser data http://genome.ucsc.edu/, Table 1). The gene length is 70.6 kb (95.2% of it accrues to introns); three transcripts are read from *CG3777* (only the longest and the shortest are shown in Figure 6D, Table 1). Within the introns of *CG3777-RB* four other genes are located: *CG13375*, *CG12470*, *Or1a* (*Odorant receptor 1a*), and partly *CG32816* (Figure 6A–D, Appendix A). According to the 4HMM model, the introns of *CG3777* throughout its length mostly contain *malachite* (42.0%) with several inclusions of *ruby* chromatin (34.6%), which correspond to neutral chromatin state (96.6%) (Figure 6B,C) [32,39]. Probes for FISH were picked from the 5’- and 3’-ends of *CG3777-RB* (red and green arrows in Figure 6D,E). The analysis showed that two probes, being separated by 69.5 kb, are detected as two distinct signals in the interbands 1A6/A7 and 1B1-2/B3-4 (Figure 6D–H). According to Bridges’ map [25], two interbands, two grey, and one black band are situated between the two probes. As not all bands from Bridges’ map are distinguishable on chromosome preparations, 1B1-2 is the only band visible in Figure 6G. It is composed of intron material that is ~67 kb in length (Figure 6).

#### 3.2.2. *CG43867*

*CG43867* is located on chromosome X in the region from 1C5 to 1D2 (according to UCSC, Table 1). The gene length is 119.7 kb (94.2% of it accrues to introns); nine transcripts are read from *CG43867* (only longest and the shortest are shown in Figure 7D, Table 1). Within the introns of *CG43867-RA*, six other genes are located: *CG14635*, *CG3713*, *CG14634*, *CG11664*, *CG11663*, and *CG3711* (Figure 7A–D, Appendix A). According to the 4HMM model, the introns of *CG43867* throughout its length mostly contain *malachite* (60.2%) with small inclusions of *ruby* chromatin (6.0%), which correspond to neutral chromatin state (92.6%) (Figure 7B,C). *CG3711* and two groups of exons on the 3’-end of *CG43867* correspond to the body of these genes and form *lazurite* chromatin (17.2%), which corresponds to neutral chromatin state (Figure 9A–D) [32,39]. The probes for FISH were picked from the 3’- and 5’-ends of *CG43867-RA* (red and green arrows in Figure 7D,E). The analysis showed that two probes, separated by 116.0 kb, are detected as two distinct signals in the interbands 1C4-5/D1-2 and 1D1-2/D3 proximal and distal of loose grey band 1D1-2 (Figure 7D–H). According to Bridges’ map [25], two interbands and one grey band, composed of intron material that is ~112 kb in length, are situated between the two probes (Figure 7).

#### 3.2.3. *br*

Previous cytogenetic and EM data show that *br* (*broad*) is located on the chromosome X in the 2B5 band (in the 2B3-2B4 region according to UCSC, Table 1) and two proximally neighboring genes *dor* (*deep orange*) and *hfw* (*halfway*) in 2B6 band [50,51,52,53]. The gene length is 70.0 kb (93.5% of it accrues to introns); 14 transcripts are read from br (only four are shown in Figure 8D, Table 1). Almost two-thirds of *br-RN* is overlaid by *Mur2B* (*Mucin related 2B*), which is located in the alternative chain (Figure 8A–D, Appendix A). Introns of *br* throughout its length mostly contain *malachite* (39.9%) with several inclusions of *ruby* chromatin (20.3%), which mostly correspond to closed (46.5%) and neutral chromatin state (41.8%), according to the 4HMM model (Figure 8B,C). Two groups of exons on the 3’-end of the gene correspond to the body of *br* and form *lazurite* chromatin (15.7%), which corresponds to neutral chromatin state (Figure 8A–D) [32,39]. *br* encodes the key protein that is necessary for the ecdysone cascade of genes and it has several activity cycles. It is induced at a high ecdysone titer (so the 2B puff appears), and then the gene is inactivated at a low ecdysone titer and such a cycle is repeated twice at the end of larval development [50,54]. This chromosome region is morphologically complex, because the bands form contacts with each other and a special puff-like structure appears (Figure 8) [4,51]. Therefore, for the preparations, we used chromosomes on the 0 h prepupa stage when the puff is already inactive. Probes for FISH were picked from the 5’-end of *br-RN* and 3’-end of *dor-RA* (red and green arrows in Figure 8D,E). The analysis showed that, when the puff is not in its active stage, two probes, separated by 88.7 kb, are detected as two distinct signals: br-I is localized in the proximal part of 2B3-4, and dor is proximally situated from it (Figure 8D–H). The complicated chromosome structure in this region does not allow for distinct identification of the 2B5 band, in which the intron part of *br* is located (Figure 8).

#### 3.2.4. *CG42666*

*CG42666* (*prage*) is located on chromosome X in the 2B9–B12 region (according to UCSC, Table 1). The gene length is 79.2 kb (96.3% of it accrues to introns); 11 transcripts are read from *CG42666* (only three are shown in Figure 9D, Table 1). Within the introns of *CG42666-RG* four other genes are located: *Adar* (*Adenosine deaminase acting on RNA*), *CG32806*, *CG14810*, and *CG14811* (Figure 9A–D, Appendix A). According to the 4HMM model, the introns of *CG42666* throughout its length mostly contains *malachite* (31.8%) with several inclusions of *aquamarine* (23.0%) and *ruby* chromatin (3.0%), which mostly correspond to open (37.6%) and neutral chromatin state (50.4%) (Figure 9B,C). Two groups of exons on the 3’-ends of *CG42666* and *Adar* correspond to the bodies of these genes and form *lazurite* chromatin (40.6%), which corresponds to open and neutral chromatin state (Figure 9A–D) [32,39]. Probes for FISH were picked from the 5’- and 3’-ends of *CG42666-RG* (red and green arrows in Figure 9D,E). The analysis showed that two probes, separated by 76.5 kb, are detected as two distinct signals in the interbands 2B7-8/B9-10 and 2B9-10/B11 proximal and distal of loose grey band 2B9-10 (Figure 9D–H). According to Bridges’ map [25], two interbands and one grey band, being composed of intron material that is ~76 kb in length, are situated between the two probes (Figure 9).

#### 3.2.5. *trol*

*trol* (*terribly reduced optic lobes*) is located on chromosome X in the 3A3–3A4 region (according to UCSC, Table 1). The gene length is 74.9 kb (81.9% of it accrues to introns); 18 transcripts are read from *trol* (only three are shown in Figure 10D, Table 1). Introns of *trol* throughout its length mostly contain *malachite* (52.0%) with two inclusions of *aquamarine* chromatin (5.2%), which mostly correspond to closed (70.7%) and neutral chromatin state (29.3%), according to the 4HMM model (Figure 10B,C). Four groups of exons on the 3’-end and in the middle of *trol* correspond to the body of this gene and form *lazurite* chromatin (42.7%), which corresponds to closed and neutral chromatin state (Figure 10A–D) [32,39]. Probes for FISH were picked from the 3’- and 5’-ends of *trol-RZ* (green and red arrows in Figure 10D,E). The analysis showed that two probes, being separated by 71.8 kb, are detected as two distinct signals in the black band 3A1-2 and in the interband 3A3/A4 (Figure 10D–H). According to Bridges’ map [25], one dense band, two interbands, and partly the adjacent black band are situated between the two probes (Figure 10).

#### 3.2.6. *sgg*

*sgg* (*shaggy*) is located on chromosome X in the 3A8–3B1 region (according to UCSC, Table 1). The gene length is 43.8 kb (93.3% of it accrues to introns); 17 transcripts are read from *sgg* (only three are shown in Figure 11D, Table 1). Introns of *sgg* throughout its length mostly contain *aquamarine* (56.9%) and *malachite* chromatin (19.7%), which correspond to open (68.5%) and neutral chromatin state (31.4%), according to the 4HMM model (Figure 11B,C). Five groups of exons on the 3’-end of the gene correspond to the body of *sgg* and form *lazurite* chromatin (21.4%), which corresponds to active chromatin state (Figure 11A–D) [32,39]. The probes for FISH were picked from the most close to the 5’- and 3’-end areas of *sgg-RA* (red and green arrows in Figure 11D,E). The analysis showed that two probes, separated by 42.4 kb, are detected as two distinct signals in the interbands 3A7/A8 and 3A10/B1 (Figure 11D–H). According to Bridges’ map [25], the whole 3A region is represented by several thin loose bands, which are very hard to distinguish with light microscopy methods. The 3A8–3A10 region comprises three neighboring bands that are composed of condensed material, which looks like a loose band that is situated between the two probes (Figure 11).

#### 3.2.7. *kirre*

*kirre* (*kin of irre*) is located between *sgg* and *dnc* (*dunce*) approximately in the 3C1–3C8 region (in the 3B4–3C7, according to UCSC, Table 1) and has a unique gene structure, but yet it has not been localized on the polytene chromosome map. The gene length is 393.7 kb (98.4% of it accrues to introns); seven transcripts are read from *kirre* (only the longest and the shortest are shown in Figure 12D, Table 1). Twenty-three other genes are located within the introns of *kirre-RF* (Figure 12A–D, Appendix A). According to the 4HMM model, introns of *kirre* throughout its length mostly contain *ruby* (50.5%) with several inclusions of *malachite* chromatin (38.6%), which mostly correspond to closed chromatin (82.0%) with small inclusions of neutral chromatin state (15.5%) (Figure 12B,C) [32,39]. The probes for FISH were picked from the *aquamarine* fragments of the gene, which match 5’-end of *kirre-RF* (kirre-I), 5’-region of *CG32795-RD* (kirre-II), 3’-end of *CG3603-RB* (kirre-III), 5’-region of *kirre-RD* (kirre-IV), and 3’-end of *kirre-RF* (kirre-V) (red and green arrows in Figure 12D,E). The analysis showed that three probes (kirre-I, kirre-III, and kirre-V) are detected as three distinct signals in the interbands 3B3-4/C1-2, 3C3-4/C5-6, and 3C7/C8. Two probes (kirre-IV, kirre-V) that are separated by 37.3 kb are detected as two distinct signals in the central and proximal parts of the 3C5-6 band. Chromosome regions between probes are represented by: thin black band 3C1 (kirre-I/kirre-II), large black band 3C2-3 (kirre-II/kirre-III) and another large black band 3C5-6 (kirre-III/kirre-IV) (Figure 12D–I). Consequently, three black bands and three interbands of the 3C1–6 region are situated within the intron of *kirre* (Figure 12).

#### 3.2.8. *dnc*

*dnc* (*dunce*) is located on chromosome X in the 3C9-3D1 region (according to UCSC, Table 1); Khoroshko et al. [45] more accurately defined its location in the region between interbands 3C7/C8–3D1/D2. The gene length is 167.3 kb (94.8% of it accrues to introns); 17 transcripts are read from *dnc* (only four are shown in Figure 13D, Table 1). Eight other genes are located within the introns of *dnc-RT* (Figure 13A–D, Appendix A). According to the 4HMM model, introns of *dnc* throughout its length mostly contain *ruby* (59.5%) with small inclusions of *malachite* chromatin (24.2%), which correspond to closed chromatin (70.5%) with small inclusions of neutral chromatin state (24.8%) (Figure 13B,C) [32,39]. The probes for FISH were picked from the 5’- and 3’-ends of *dnc-RT* and from 5’-end of *dnc-RJ*, in all the cases matching with fragments of *aquamarine* chromatin (black arrows in Figure 13D,E). The analysis showed that three probes, which are separated by 78.5 kb and 85.9 kb, are detected as three distinct signals in the interbands 3C7/C8, 3C9-10/C11-12, and 3D1,2/D3,4 (Figure 13D–I). The distinct location of the dnc-I and dnc-II probes is most visible on the stretched chromosomes preparations (Figure 13F). Seven interbands and six grey bands are situated between the dnc-I and dnc-II probes (Figure 13), according to Bridges’ map [25].

#### 3.2.9. *Nrg*

*Nrg* (*Neuroglian*) is located on chromosome X in the 7F2–7F4 region (according to UCSC, Table 1). The gene length is 37.7 kb (80.4% accrues to introns); nine transcripts are read from *Nrg* (only the longest is shown in Figure 14D, Table 1). One gene is located within the introns of *Nrg-RE*–*PIP82* (Figure 14A–D, Appendix A). According to the 4HMM model, the introns of *Nrg* throughout its length mostly contain *malachite* (60.3%) with two small inclusions of *ruby* chromatin (7.4%), which correspond to neutral chromatin state (93.1%) (Figure 14B,C). The group of exons on the 3’-end of the gene corresponds to the body of *Nrg* and form *lazurite* chromatin (24.8%), which corresponds to neutral chromatin state (Figure 14A–D) [32,39]. Probes for FISH were picked from the 5’- and 3’-ends of *Nrg-RE* (red and green arrows in Figure 14D,E). The analysis showed that two probes, being separated by 36.3 kb, are detected as two distinct signals in the interbands 7F1-2/F3-4 and 7F3-4/F5-6 proximal and distal of the loose grey band 7F3-4 (Figure 14D–H). According to Bridges’ map [25], two interbands and one grey band, being composed of intron material that is ~36 kb in length, are situated between the two probes (Figure 14).

#### 3.2.10. *dlg1*

*dlg1* (*discs large 1*) is located on chromosome X in the 10B6–10B11 region (according to UCSC, Table 1); Zhimulev et al. [46] have preliminarily localized it in the 10B8–1011 region. The gene length is 40.1 kb (81.9% of it accrues to introns); 21 transcripts are read from *dlg1* (only three are shown in Figure 15D, Table 1). Half of *dlg1* introns contain *malachite* (41.3%) with one inclusion of *ruby* chromatin (3.4%), which mostly correspond to neutral chromatin state (48.3%), according to the 4HMM model (Figure 15B,C). Four groups of exons on the 3’-end of the gene correspond to the body of *dlg1* and form *lazurite* chromatin (39.4%), which corresponds to active chromatin state (51.6%) (Figure 15A–D) [32,39]. The probes for FISH were picked from *Tim8* and *Or10a* (*Odorant receptor 10a*) genes, and from 5’-end of *dlg1-RT* (red and green arrows in Figure 15D,E). The analysis showed that three probes, which are separated by 20.5 kb and 26.1 kb, are detected as three distinct signals in the interbands 10B7/B8-9, 10B8-9/10-11, and 10B10-11/B12 (Figure 15D–H). Three interbands and two grey bands are situated between the Tim8 and Or10a probes, according to Bridges’ map [25] (Figure 15).

#### 3.2.11. *EcR*

*EcR* (*Ecdysone receptor*) is located on chromosome 2R in the 42A9-42A12 (according to UCSC, Table 1). The gene length is 78.6 kb (93.5% of it accrues to introns); six transcripts are read from *EcR* (only the longest and the shortest and are shown in Figure 16D, Table 1). Within the introns of *EcR-RA* two other genes are located: *CG14589*, *CR43904*, and also 13 RNA transport genes, which we combined into four groups (Figure 1A–D, Appendix A). The introns of *EcR* throughout its length mostly contain *malachite* (55.2%), with four inclusions of *ruby* (7.3%) and *aquamarine* chromatin (24.1%), which correspond to open (68.8%) and neutral chromatin state (31.1%), according to the 4HMM model (Figure 16B,C). The group of exons on the 3’-end of the gene correspond to the body of *EcR* and form *lazurite* chromatin (4.8%), which corresponds to neutral chromatin state (Figure 16A–D) [32,39]. The probes for FISH were picked from the area most close to 3’-end and from 5’-end of *EcR-RA* (green and red arrows in Figure 16D,E). The analysis showed that two probes, separated by 79.6 kb, are detected as two distinct signals in the 42A12-A16 region (Figure 14D–H). Two probes bound a small section of compact material, presumably the 42A15 band, which derives from the intron (black arrow in Figure 16F–H). In this grey decondensed band between the two probes, three clusters of transport RNA genes are located (Figure 16D,E).

#### 3.2.12. *Hr46*

*Hr46* (*Hormone receptor-like in 46*, *Hormone receptor 3*) is located on chromosome 2R in the 46F5–46F7 region (according to UCSC, Table 1). The gene length is 31.8 kb (86.7% of it accrues to introns); seven transcripts are read from *Hr46* (only three are shown in Figure 17D, Table 1). One gene is located within the introns of *Hr46-RD*–*CG12912* (Figure 17A–D, Appendix A). Introns of *Hr46* throughout its length mostly contain *ruby* (51.0%), with several inclusions of *malachite* chromatin (39.5%), which mostly correspond to closed chromatin (66.4%) with inclusions of neutral chromatin state (31.4%), according to the 4HMM model (Figure 17B,C) [32,39]. The probes for FISH were picked from the 3’- and 5’-ends of *Hr46-RD* (green and red arrows in Figure 17D,E). The analysis showed that two probes, separated by 30.6 kb, are detected as two distinct signals in the black band 46F5-6 and in the interband 46F7/F8 (Figure 17D–H). *Hr46*, a short and densely packed gene, which is inactive in most tissues, does not allow for clearly seeing the inactive intron material between the probes (Figure 17).

#### 3.2.13. *Eip74EF*

*Eip74EF* (*Ecdysone-induced protein 74EF*) is located on chromosome 3L in the region of ecdysone-induced puff, which, according to EM data, is formed by two grey loose bands 74E1 and 74E2 [56]. The gene length is 59.1 kb (89.8% of it accrues to introns); five transcripts are read from *Eip74-EF* (only the longest and the shortest are shown in Figure 18D, Table 1). Introns of *Eip74EF* throughout its length mostly contain *malachite* (60.7%) with three inclusions of *ruby* (7.7%) and *aquamarine* chromatin (27.4%), which correspond to open (83.2%) and neutral chromatin state (14.3%), according to the 4HMM model (Figure 18B,C). The group of exons on the 3’-end of the gene correspond to the body of *Eip74EF* and form *lazurite* chromatin (4.0%), which corresponds to active chromatin state (Figure 18A–D) [32,39]. The probes for FISH were picked from the 3’-end of *Eip74EF-RD* and 5’-end of *Eip74EF-RA* (green and red arrows in Figure 18E). The analysis showed that two probes, which are separated by 60.2 kb, are detected as two distinct signals in the interbands 74D5/E1-2 and 74E5/F1 (Figure 18D–I). The distinct location of the probes is most visible on the stretched chromosomes preparations (Figure 18H). According to Bridges’ map [57], five interbands and four grey bands are situated between the probes (Figure 18).

#### 3.2.14. *Eip75B*

*Eip75B* (*Ecdysone-induced protein 75B*) is located on chromosome 3L in the region of ecdysone-induced puff, which, according to EM data, is formed by five simultaneously decondensed bands 75B1-2, 75B3, 75B4, 75B5, and 75B6-7 of Bridges map [56]. The gene length is 113.7 kb (94.8% of it accrues to introns); six transcripts are read from *Eip75B* (only four are shown in Figure 19D, Table 1). Within the introns of Eip75B-RF, two other genes are located: *CG32192*, *CG42393*, and *snoRNA:Me28S-A30* (Figure 19A–D, Appendix A). According to the 4HMM model, introns of Eip75B throughout its length mostly contain *malachite* (46.9%) with several inclusions of *ruby* (11.9%) and *aquamarine* chromatin (30.9%), which correspond to open (48.9%) and neutral chromatin state (49.2%) (Figure 19B,C). The group of exons on the 3’-end of the gene correspond to the body of *Eip75B* and form *lazurite* chromatin (2.4%), which corresponds to active chromatin state (Figure 19A–D) [32,39]. The probes for FISH were picked from the 3’- and 5’-ends of *Eip75B-RF* (green and red arrows in Figure 19D,E). The analysis showed that two probes, which are separated by 113.2 kb, are detected as two distinct signals on the edge of black band 75B1-2 and in the loose band 75B6-7 (Figure 19D–H). Four interbands and five bands are situated between the probes, according to Bridges’ map [57] (Figure 19).

#### 3.2.15. *Eip78C*

*Eip78C* (*Ecdysone-induced protein 78C*) is located on chromosome 3L in the region of ecdysone-induced puff, which, according to EM data, is formed by three grey loose bands 78C4, 78C5-6 and 78C6, not always being visible under the light microscope [56]. The gene length is 39.5 kb (86.8% of it accrues to introns); four transcripts are read from *Eip78C* (only the longest and the shortest are shown in Figure 20D, Table 1). One gene is located within the introns of *Eip78C-RA*–*CG43218* (Figure 20A–D, Appendix A). Introns of *Eip78C* throughout its length mostly contain *malachite* (39.4%) with several inclusions of *ruby* chromatin (25.2%), which correspond to closed (59.4%) and neutral chromatin state (28.5%), according to the 4HMM model (Figure 20B,C) [32,39]. Probes for FISH were picked from the area most close to 5’-end and from 3’-end of *Eip78C-RA* (red and green arrows in Figure 20D,E). The analysis showed that two probes, separated by 38.5 kb, are detected as two distinct signals in the interbands 78C3/C4 and 78C5-6/C7 (Figure 20D–H). Three interbands and two bands are situated between the probes, according to Bridges’ map [57] (Figure 20).

## 4. Discussion

The present study on the distribution of introns in *Drosophila* genes showed that they are mainly detected in the bodies of housekeeping genes and their length mostly does not exceed 2 kb. The introns are predominantly distributed in *lazurite* chromatin, which corresponds to gene bodies, and also in adjacent *aquamarine* and *ruby* (Figure 2A). In gene bodies, the largest total intron lengths are localized in *ruby* chromatin along with developmental genes. In almost all long genes with the total intron length over 95%, a significant part of introns is covered with *malachite* chromatin. Apparently, the gene length increases with increasing intron length, but, at the same time, the proportion of very long genes drops sharply in genes with a length over 50 kb, and especially over 100 kb; their number is no more than 70 (approximately 0.5% of the total number of genes in the *Drosophila* genome).

As expected, the distribution of genes and chromatin states corresponds to patterns that were discovered in another study [39]; both the promoters of 12 longest transcripts and the most of alternative transcripts were located in fragments of open *aquamarine* chromatin and at the chromosome level in interbands, which indicates that these genes pertain to housekeeping. Coding parts of the genes are located in *lazurite*. The total length of *malachite* chromatin (mainly introns) in the majority of selected genes comprises 41.9% of these genes length; *ruby* totals 28.3% (Figure 5C). 5’-ends of *br*, *Hr46* and *Eip78C* genes are not located in *aquamarine* chromatin and, subsequently, not in the interbands; the data on gene functioning in target tissues does not indicate that they are involved in housekeeping (Appendix A). *Hr46* is a tissue-specific gene, because it only functions in the imago brain and *Eip78C* is only active in the larval fat body (Appendix A). The *br* gene is peculiar, because its alternative promoters are found in *aquamarine* and apparently in the 2B3-4/B5 interband. It should be noted that this gene is located in the most difficult for cytological mapping 2B region [4,50,51] and, therefore, it is difficult to draw conclusions regarding the gene localization. The predominant states of chromatin in these three genes are *ruby* and *malachite*, as well as the neutral and closed chromatin condensation states [32] (Figure 8C, Figure 17C, Figure 20C).

Figure 21, Figure 22 and Figure 23 show how genes that are composed of introns form bands and interbands. In the most common case, bands that formed from intron material are located between the promoter at the 5’-end and the intergenic region (interband) at the 3’-end of the gene (Figure 21). A band of this kind is genetically and molecularly heterogeneous, since it contains the intron material (*ruby* and *malachite*) and exons of the gene (*lazurite*) (Figure 21A). Within the noncoding intron of the predominant long “main” gene, one or several differently directed genes are located; in some cases, there are no such genes (Figure 21B,C). This type includes bands that formed from the introns in *CG3777* (1A8–1B1-2), *CG43867* (1D1-2), *CG42666* (2B9-10), *trol* (3A1-3), *sgg* (3A9-10), *Nrg* (7F2-4), *EcR* (42A14), *Eip74EF* (bands in 74EF region), and *Eip75B* (bands in 75B region).

*br*, *Hr46*, and *Eip78C* are partly similar to the type that is described above; as developmental genes they do not start in *aquamarine*. However, there is a lot of material in the gene body and it forms a dense grey clot, which is not a band, but it easily separates the signals of the gene start and end. Therefore, the material between the gene start and end looks grey, loose, and semi-condensed, and it is not a usual band, as it is genetically heterogeneous and contains both protein-coding and intron material.

In some cases, bands are formed from intron material, as shown in the *dlg1* and *dnc* genes. The promoter of the *dlg1 gene* and the alternative promoter (situated in the middle of the DNA fragment occupied by the gene) are both located in *aquamarine*, on cytological map matching with interbands (Figure 15, Figure 22). A ~20 kb intron covered with *malachite* and one *ruby* inclusion is situated between the distal and middle promoters. It forms the 10B8-9 band that only contains intron DNA of the *dlg1* gene (Figure 15, Figure 22A,B). The second 10B10-11 band is formed from the material located between the middle promoter and the 3’-end of the gene; it comprises introns and protein-coding parts of the gene (Figure 15, Figure 22B). Thus, *dlg1* occupies five chromosome structures of the polytene chromosome X, i.e. two bands and three interbands.

The *dnc* gene is as long as 167.3 kb and almost 95% of it accrues to introns (Table 1). The FISH analysis showed that the dnc-I probe (the promoter region of the longest transcript *dnc-RT*) is located in the 3C7/C8 interband; dnc-II probe (3’-end of *dnc-RT*) is located in 3D1/D2 interband (Figure 13). Thus, according to Bridges’ map [25], *dnc* occupies five bands and six interbands (as this complex region comprises thin and almost indistinguishable bands, we hold on to the number of bands on the map). The dnc-III probe and the second promoter of the *dnc* gene, respectively, are located in the interband 3C9-10/C11-12 (Figure 13). The region between the dnc-III and dnc-II probes (~86 kb in length) is quite homogeneous; it contains two very thin bands 3C11-12 (supposedly represented by intron material) and two loosened large bands 3D1 and 3D2, in which the material of the 3’-end part of the gene is apparently located. The short genes (like *CG14265*, *ng1*, *ng3*, etc.) that are located within introns of *dnc* are active predominantly in larval salivary gland cells (see Appendix A).

In all of these cases, bands that formed from intron material are mainly characterized by the neutral level of chromatin condensation, *malachite* coverage (in some cases with *ruby* and/or *lazurite* inclusions); on the chromosomes they correspond to grey loose decompacted bands. For example, 3A2-3 band looks black and contains *ruby*; 3A8-9 band is almost completely decondensed, and chromatin is open for transcription, according to both 4HMM and 3CM models [32,39]. The decompacted chromatin state usually corresponds to a certain level of transcription: in 0 h prepupa grey bands 1D1-2 (*CG43867*) and 3A9-10 (*sgg*) demonstrate high transcriptional activity visualized by ^3^H-uridine incorporation [58]; according to the recent modENCODE data, an active accumulation of RNA pol II is registered in these regions. 

The localization of shorter genes within the introns of the predominant long “main” genes is an interesting feature. The data analysis showed that the studied long genes themselves and the shorter genes lying within its introns function in completely different tissues (Appendix A). For example, the “main” gene *CG43867* in the 1D1-2 band is active in an extensive set of tissues on many developmental stages, while the short genes function in a very limited number of cells or organs that are related to male fertility. *EcR* is a moderately housekeeping gene; two genes affecting male fertility and at least four clusters of transport RNA genes are located within its introns (see other examples in Appendix A).

The *kirre* gene appears to be of particular interest. In polytene chromosomes, this gene has even more complex genetic organization (Figure 23). It is one of the longest genes in the *Drosophila* genome, as it occupies 393.7 kb from the promoter (probe kirre-I) to the 3’-end (probe kirre-V), and 98.4% of its length accrues to introns (Table 1). Almost all of material of the gene looks like compact, transcriptionally inactive, and late replicating polytene chromosome bands [43,56,59,60]. At chromatin level (Figure 12), this material is also highly compact [35]. Within the introns of *kirre* 23 other genes are located (Figure 12D, Appendix A); some of them demonstrate a moderate level of housekeeping activity, and the rest are developmental genes (Figure 23B,C).

Compact material, within which *kirre* and other short genes are located, is divided by four interbands in locations of kirre-I, kirre-II, kirre-III, and kirre-V probes, where 5’-ends of four housekeeping genes are also localized. Probe kirre-IV is also located in *aquamarine*, but it was detected on the proximal edge of the 3C5-6 band (Figure 12). The probe kirre-I is localized alongside with the promoter of *kirre*, whole *Syx4* gene, as well as the proteins and genome elements characteristic of interbands which are situated within the ~6.5 kb *aquamarine* fragment (corresponds to the 3B4-5/C1 interband) (Appendix A). The kirre-II probe is localized in *aquamarine* fragment (corresponds to the 3C1/C2-3 interband), where the promoter of the moderately active housekeeping gene *CG32795* is also located (Appendix A). Similarly, the probe kirre-III is localized in *aquamarine* fragment (corresponds to 3C2-3/C5-6 interband), where the moderately active whole housekeeping gene *CG3603* is situated (Appendix A, Appendix A). Finally, the probe kirre-V is localized in *aquamarine* fragment (corresponds to the 3C5-6/C7 interband), together with the 3’-end of *kirre* and the promoter of the *Notch* gene (Appendix A) [61,62].

Thus, the locus of the *kirre* gene contains 23 shorter mostly inactive genes, which, together with *kirre*, form compact band material. In those housekeeping genes where transcription occurs, the promoters are active and interbands are being formed (Figure 23). Therefore, interbands are formed throughout the length of the extended “main” gene due to the random arrangement of moderately active, but functionally not related, housekeeping genes.

The essential result of this study is the discovery of a novel principle of gene arrangement in polytene chromosomes. In the past, polytene chromosome bands were considered to be sites where only one gene or promoter part and gene body can be located; citologically, they could be found in band/interband, interband/band structure or in a black bands group of functionally not related genes (for details, see [19]). The present study shows that genes containing extended introns appear to occupy long chromosome regions composed of band and interband series. The gene material depending on its chromosomal composition is shown to be arranged in two ways: the coding parts of genes are located in grey bands (*lazurite* chromatin) and intronic material (*malachite* and *ruby* chromatin) form bands, which can contain either both exons and introns or intron material only. The promoter regions of such genes are located in *aquamarine* fragments (interbands) and the gene bodies occupy long DNA sequences, which comprise various chromatin states and different chromosome structures.

It is remarkable that the long genes are involved in ecdysone hormone signalling cascade, which plays the key role during development. These genes demonstrate very complex molecular organization; alternative promoters and exons assemble mRNA from different gene fragments that are active at different developmental stages and demonstrate different genetic significance [52,53,63,64]. We might speculate that this genetic complexity might correlate with cytological picture; however, this kind of analysis might be the subject of future studies.

## Figures and Tables

**Figure 1 genes-11-00417-f001:**
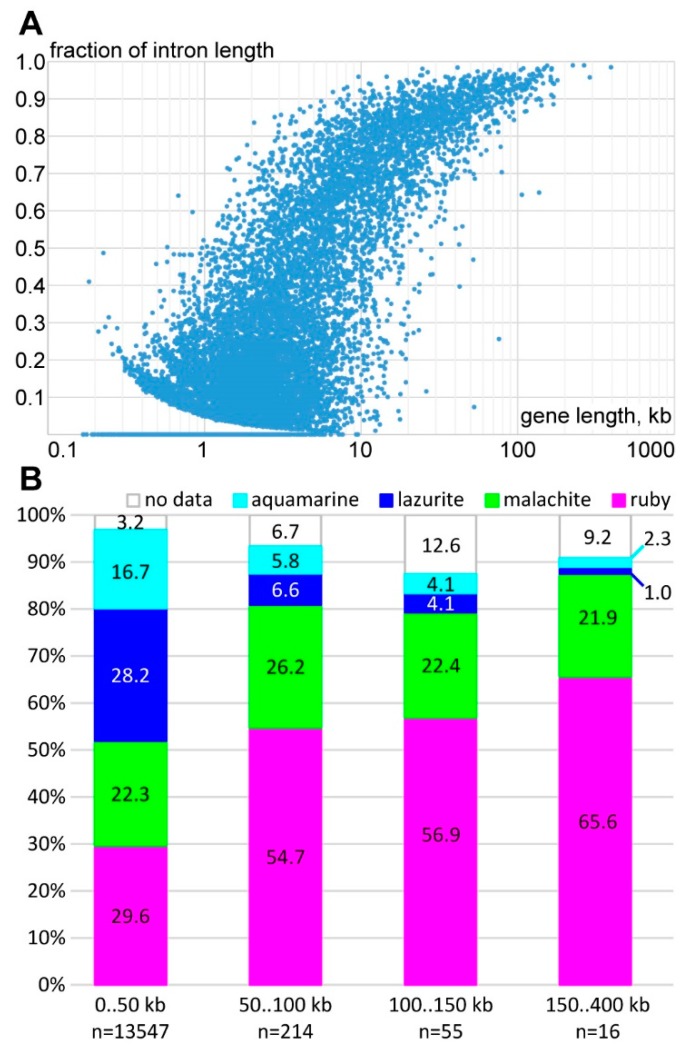
The distribution of introns and different chromatin states according to 4HMM model [39]. (**A**) fraction of intron length according to the gene length in kb; and, (**B**) distribution of four chromatin states in genes with length <2, 2–50, 50–100, and >100 kb).

**Figure 2 genes-11-00417-f002:**
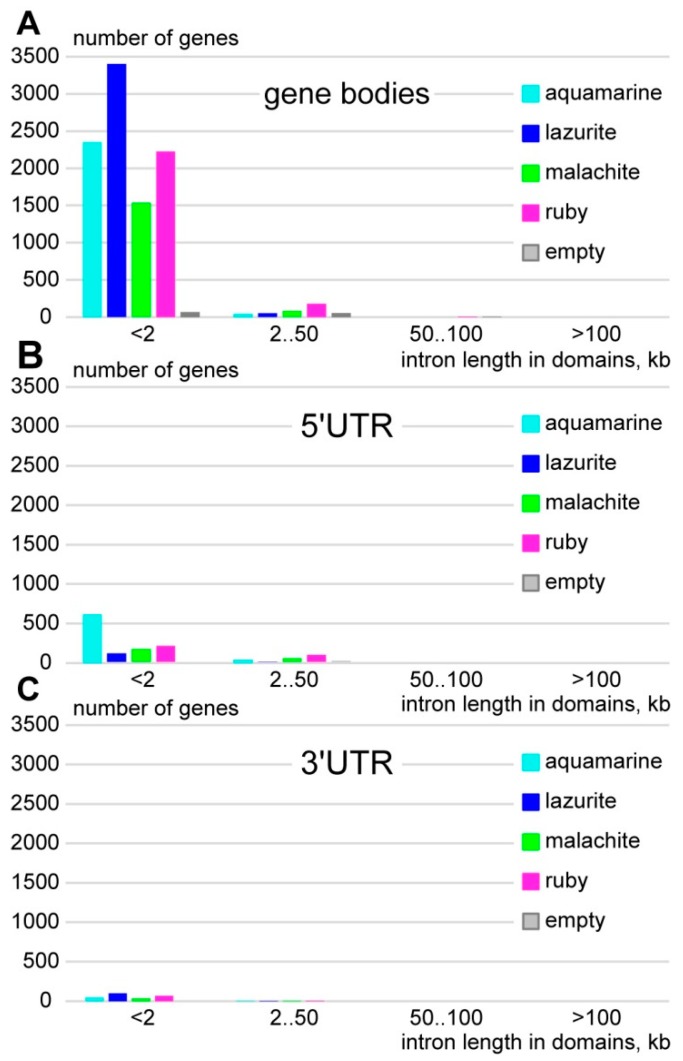
Distribution of the intron fractions and four chromatin states according to gene length in the *Drosophila* genome. *X*-axis represents gene length (bp); *Y*-axis represents the proportion of introns in the gene; black line represents introns on which 4HMM model does not provide any data. (**A**) intron fractions in gene bodies; (**B**) intron fractions in 5’UTR; and, (**C**) intron fractions in 3’UTR.

**Figure 3 genes-11-00417-f003:**
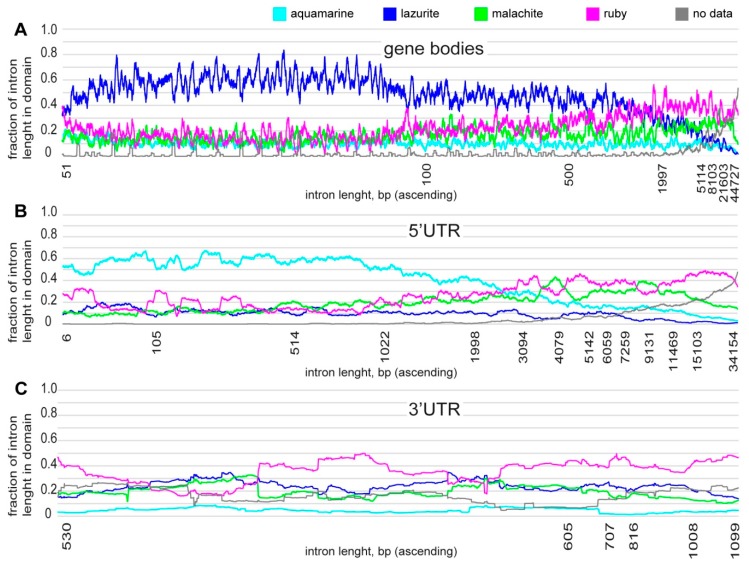
Intron fraction overlapping with four-chromatin states model [39] according to intron length (the calculations were carried out using moving average on the number of introns equaling 500; black line presents genome regions for which data on chromatin states is absent). (**A**) distribution for gene bodies; (**B**) distribution for 5’UTR; and, (**C**) distribution for 3’UTR.

**Figure 4 genes-11-00417-f004:**
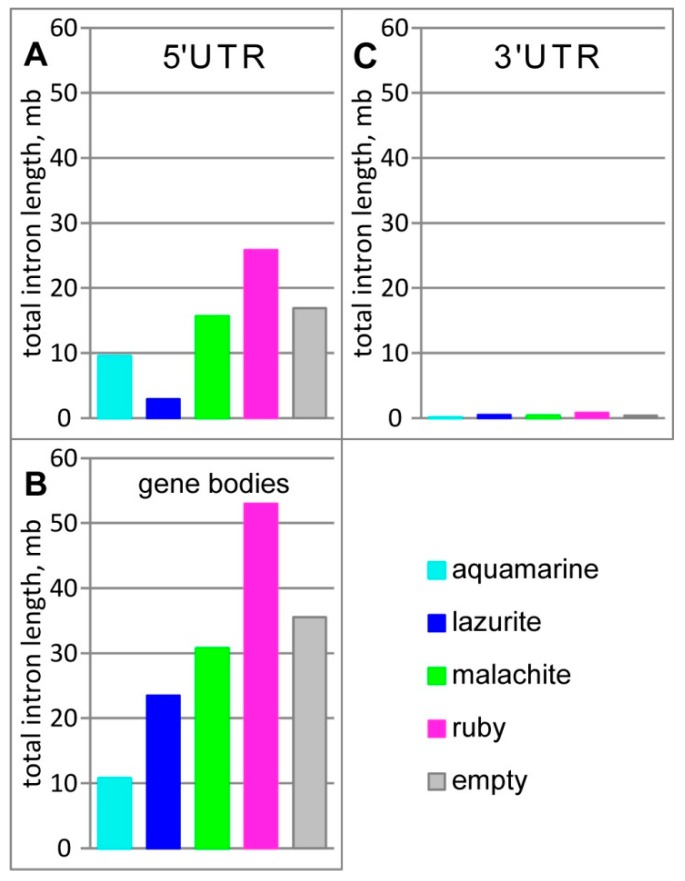
Summed intron length (Mb) overlapping with four-chromatin states model [39]. (**A**) in 5’UTR; (**B**) in gene bodies; and, (**C**) in 3’UTR.

**Figure 5 genes-11-00417-f005:**
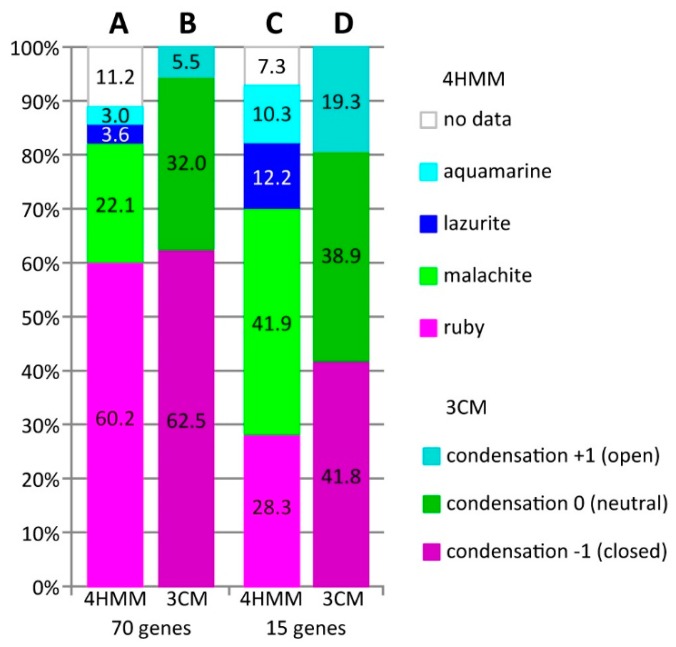
Summed length of four-chromatin states model [39] and three chromatin condensation level model [32] in 70 and 15 studied genes. (**A**) distribution of chromatin states in 4HMM model in 70 genes (%); (**B**) distribution of chromatin condensation states in 3CM model in 70 genes (%); (**C**) distribution of chromatin states in 4HMM model in 15 genes (%); and, (**D**) distribution of chromatin condensation states in 3CM model in 15 genes (%).

**Figure 6 genes-11-00417-f006:**
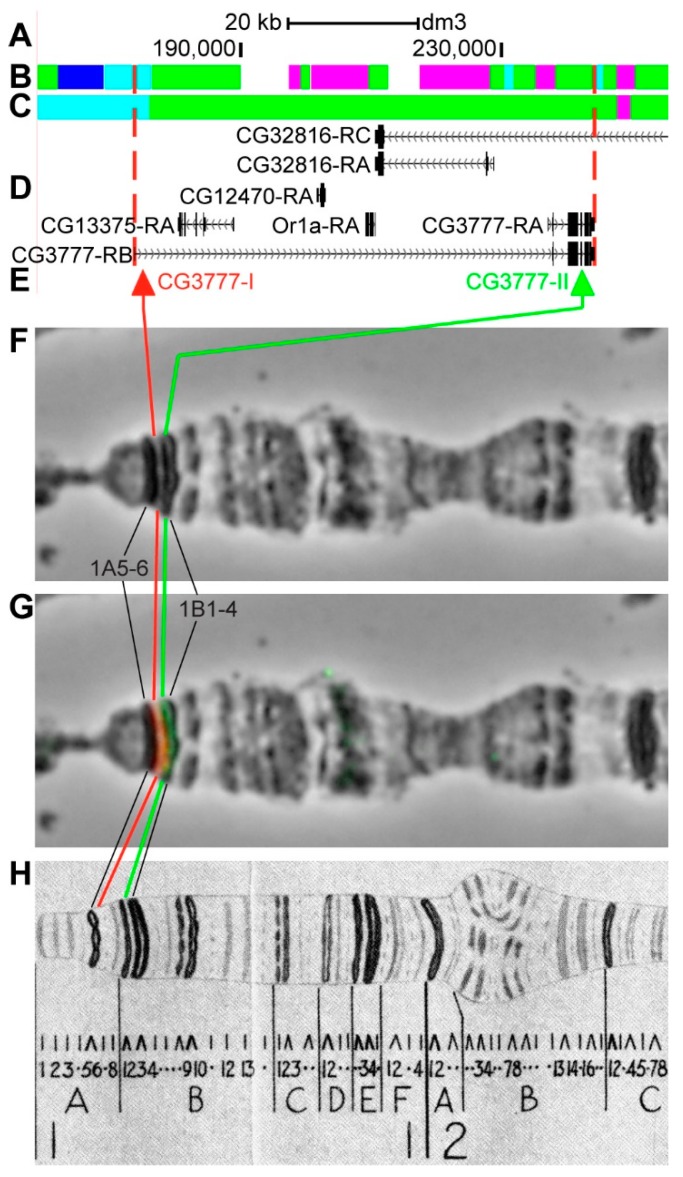
Localization of the *CG3777* gene in the 1AB region (the gene is bound by red dotted lines). (**A**) the scale (kb) and genomic coordinates (bp); (**B**) the four chromatin-state model [39]; (**C**) the chromatin condensation level model [32]; (**D**) genes location (arrow lines show the gene orientation and black bars indicate the location of gene exons); (**E**) probes location; (**F**) phase-contrast micrograph of the region; (**G**) combined fluorescence in situ hybridization (FISH) signals; and, (**H**) C. Bridges’ chromosome map [25].

**Figure 7 genes-11-00417-f007:**
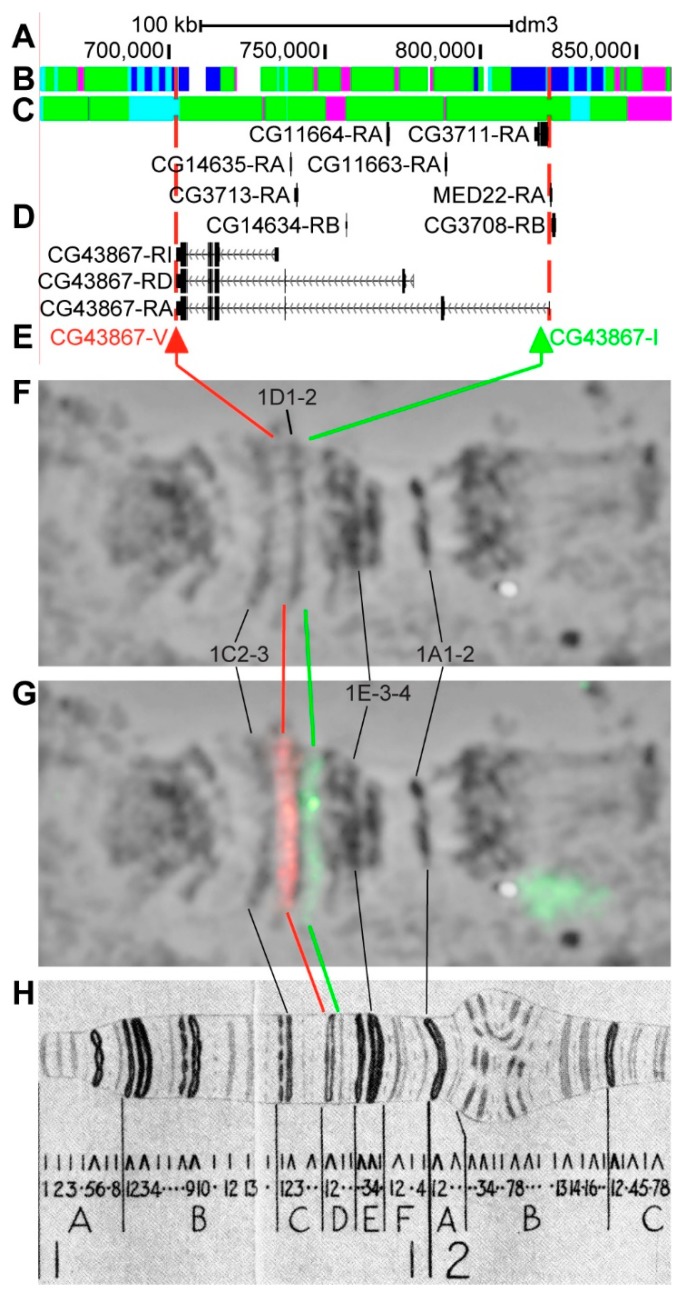
Localization of the *CG43867* gene in the 1CD region (the gene is bound by red dotted lines). (**A**) the scale (kb) and genomic coordinates (bp); (**B**) the four chromatin-state model [39]; (**C**) the chromatin condensation level model [32]; (**D**) genes location (arrow lines show the gene orientation and black bars indicate the location of gene exons); (**E**) probes location; (**F**) phase-contrast micrograph of the region; (**G**) combined FISH signals; and, (**H**) C. Bridges’ chromosome map [25].

**Figure 8 genes-11-00417-f008:**
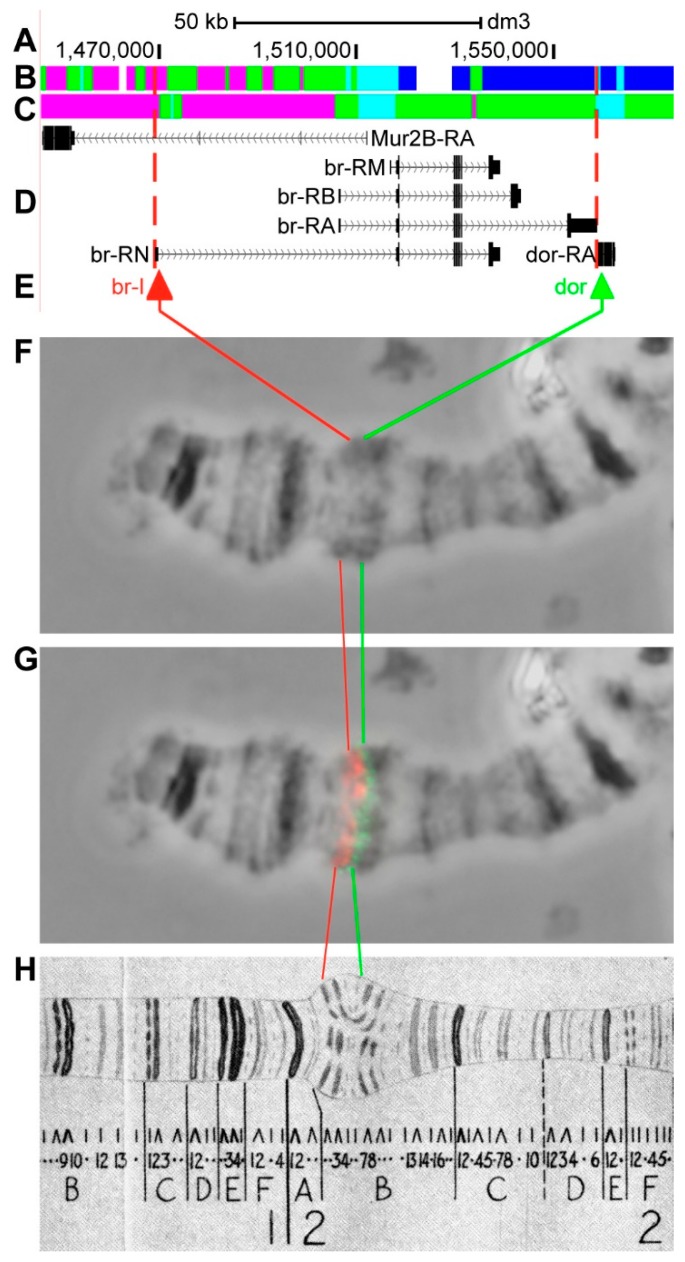
Localization of the *br* gene in the 2B region (the gene is bound by red dotted lines). (**A**) the scale (kb) and genomic coordinates (bp); (**B**) the four chromatin-state model [39]; (**C**) the chromatin condensation level model [32]; (**D**) genes location (arrow lines show the gene orientation and black bars indicate the location of gene exons); (**E**) probes location; (**F**) phase-contrast micrograph of the region; (**G**) combined FISH signals; and, (**H**) C. Bridges’ chromosome map [25].

**Figure 9 genes-11-00417-f009:**
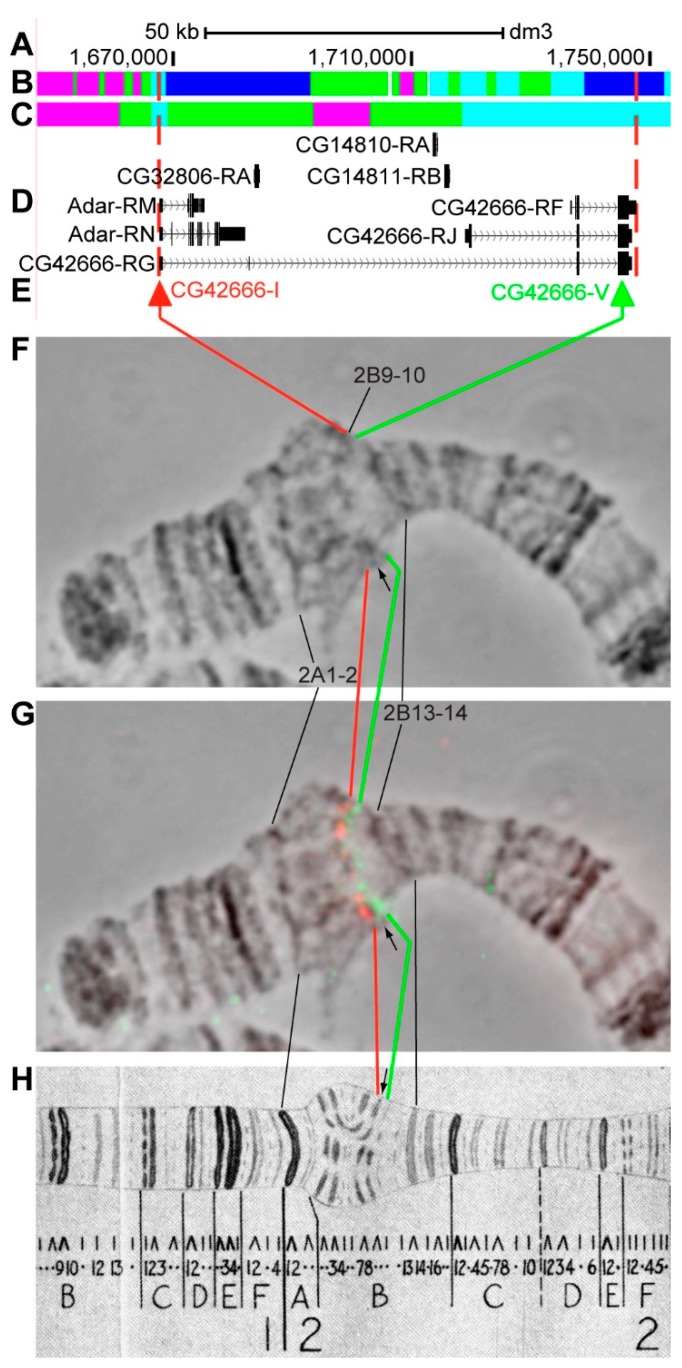
Localization of the *CG42666* gene in the 2B region (the gene is bound by red dotted lines). (**A**) the scale (kb) and genomic coordinates (bp); (**B**) the four chromatin-state model [39]; (**C**) the chromatin condensation level model [32]; (**D**) genes location (arrow lines show the gene orientation and black bars indicate the location of gene exons); (**E**) probes location; (**F**) phase-contrast micrograph of the region; (**G**) combined FISH signals; and, (**H**) C. Bridges’ chromosome map [25].

**Figure 10 genes-11-00417-f010:**
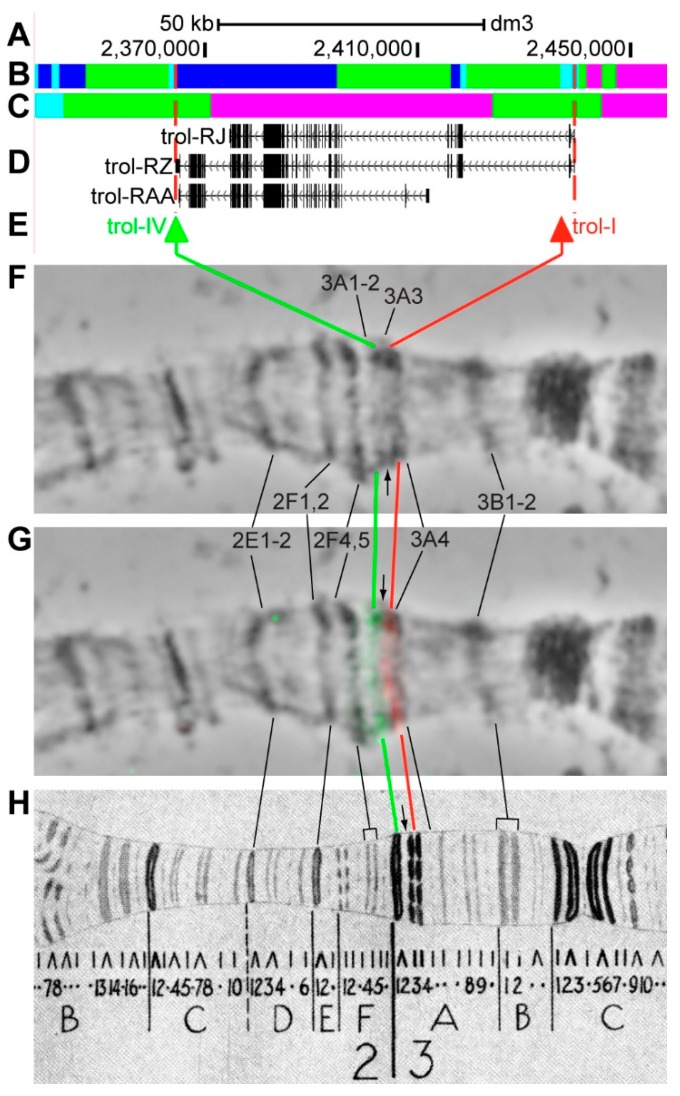
Localization of the *trol* gene in the 3A1-4 region (the gene is bound by red dotted lines). (**A**) the scale (kb) and genomic coordinates (bp); (**B**) the four chromatin-state model [39]; (**C**) the chromatin condensation level model [32]; (**D**) genes location (arrow lines show the gene orientation and black bars indicate the location of gene exons); (**E**) probes location; (**F**) phase-contrast micrograph of the region; (**G**) combined FISH signals; and, (**H**) C. Bridges’ chromosome map [25].

**Figure 11 genes-11-00417-f011:**
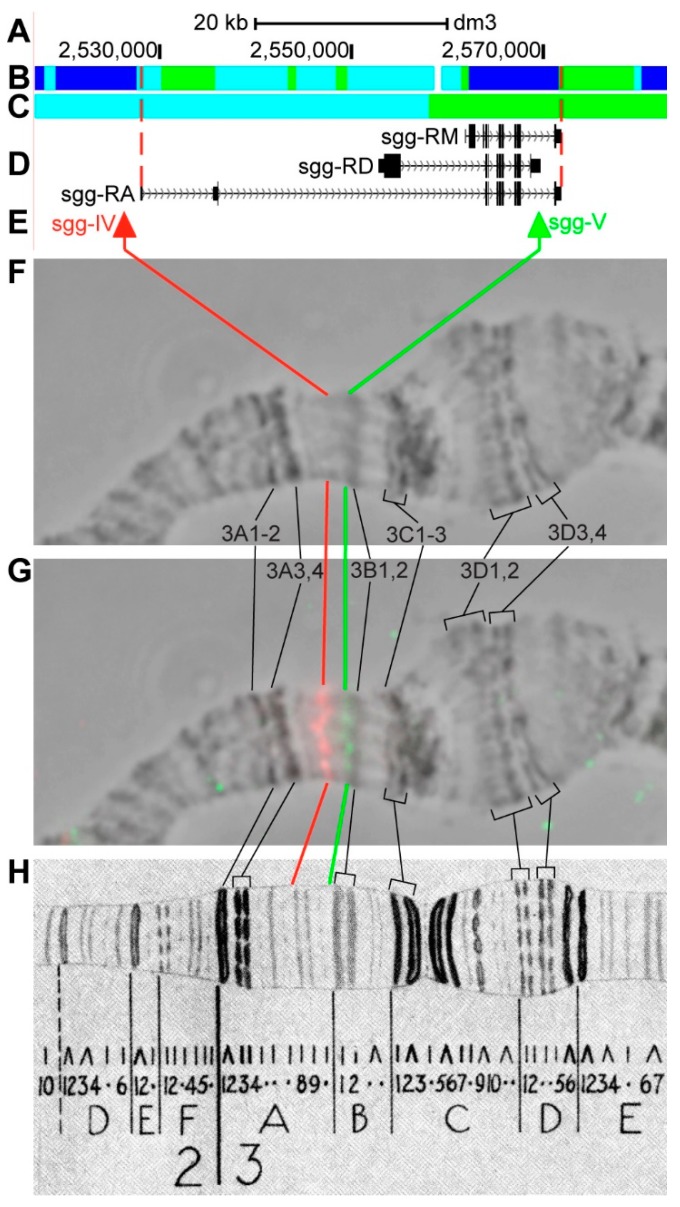
Localization of the *sgg* gene in the 3AB region (the gene is bound by red dotted lines). (**A**) the scale (kb) and genomic coordinates (bp); (**B**) the four chromatin-state model [39]; (**C**) the chromatin condensation level model [32]; (**D**) genes location (arrow lines show the gene orientation and black bars indicate the location of gene exons); (**E**) probes location; (**F**) phase-contrast micrograph of the region; (**G**) combined FISH signals; and, (**H**) C. Bridges’ chromosome map [25].

**Figure 12 genes-11-00417-f012:**
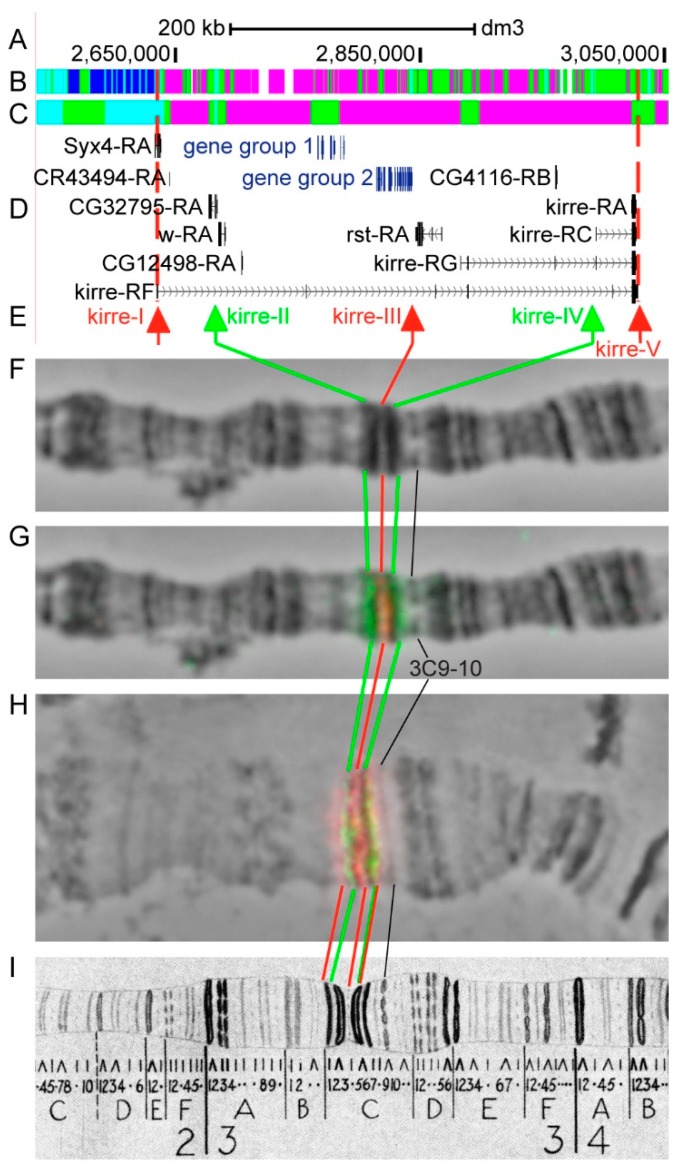
Localization of the *kirre* gene in the 3C region (the gene is bound by red dotted lines). (**A**) the scale (kb) and genomic coordinates (bp); (**B**) the four chromatin-state model [39]; (**C**) the chromatin condensation level model [32]; (**D**) genes location (arrow lines show the gene orientation and black bars indicate the location of gene exons); (**E**) probes location; (**F**) phase-contrast micrograph of the region; (**G**) combined FISH signals on the stretched chromosome preparation for kirre-II, kirre-III, and kirre-IV probes; (**H**) combined FISH signals for kirre-I, kirre-II, kirre-III, kirre-IV, and kirre-V probes; and, (**I**) C. Bridges’ chromosome map [25]. Gene group 1 is represented by: *CG14416*, *CG14417*, *CG14418*, *CG14419*, *CG3526*; gene group 2 is represented by: *CG3588*, *CG14424*, *CG32793*, *CG3592*, *CG3598*, *CG14420*, *CG14421*, *CG14422*, *CG14423*, *CG17959*, *CG3603*.

**Figure 13 genes-11-00417-f013:**
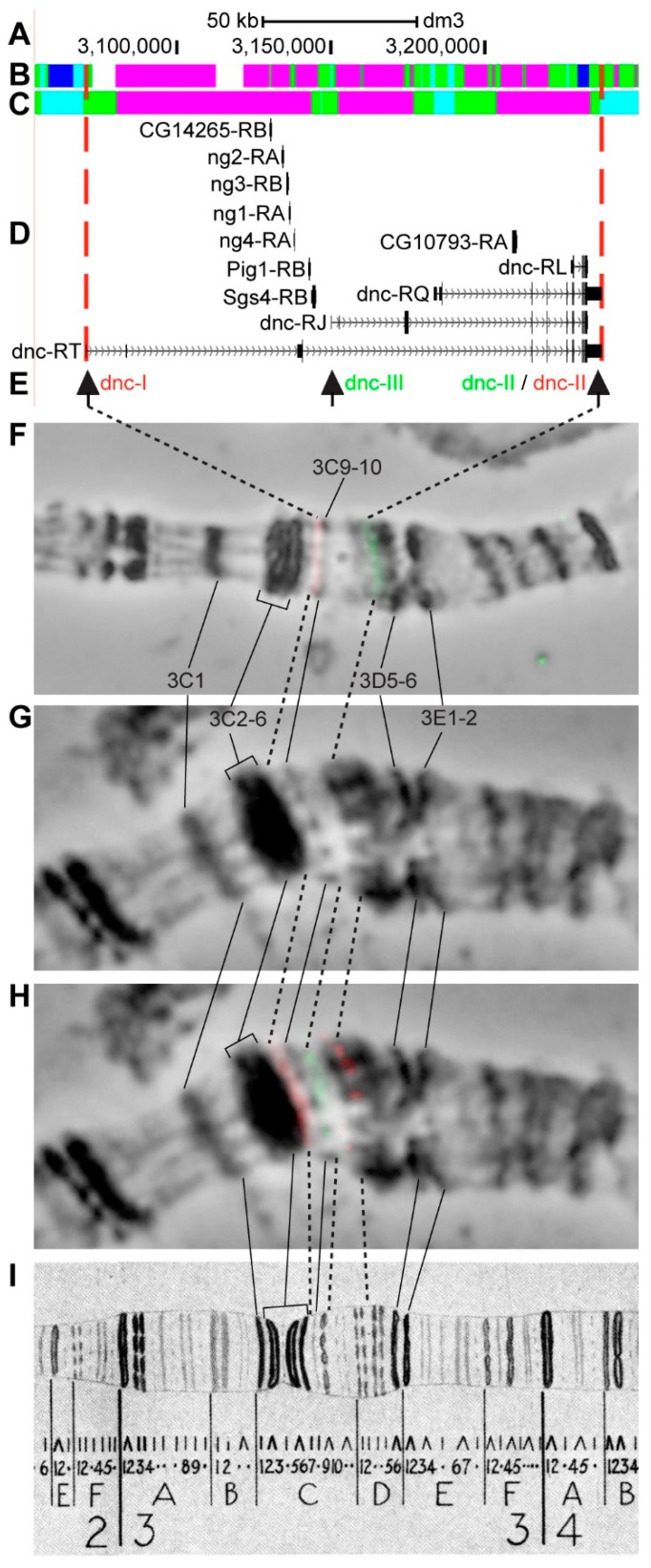
Localization of the *dnc* gene in the 3CD region (on the molecular map the gene is bound by red dotted lines). (**A**) the scale (kb) and genomic coordinates (bp); (**B**) the four chromatin-state model [39]; (**C**) the chromatin condensation level model [32]; (**D**) genes location (arrow lines show the gene orientation and black bars indicate the location of gene exons); (**E**) probes location; (**F**) combined FISH signals on the stretched chromosome preparation; (**G**) phase-contrast micrograph of the region; (**H**) combined FISH signals; and, (**I**) C. Bridges’ chromosome map [25].

**Figure 14 genes-11-00417-f014:**
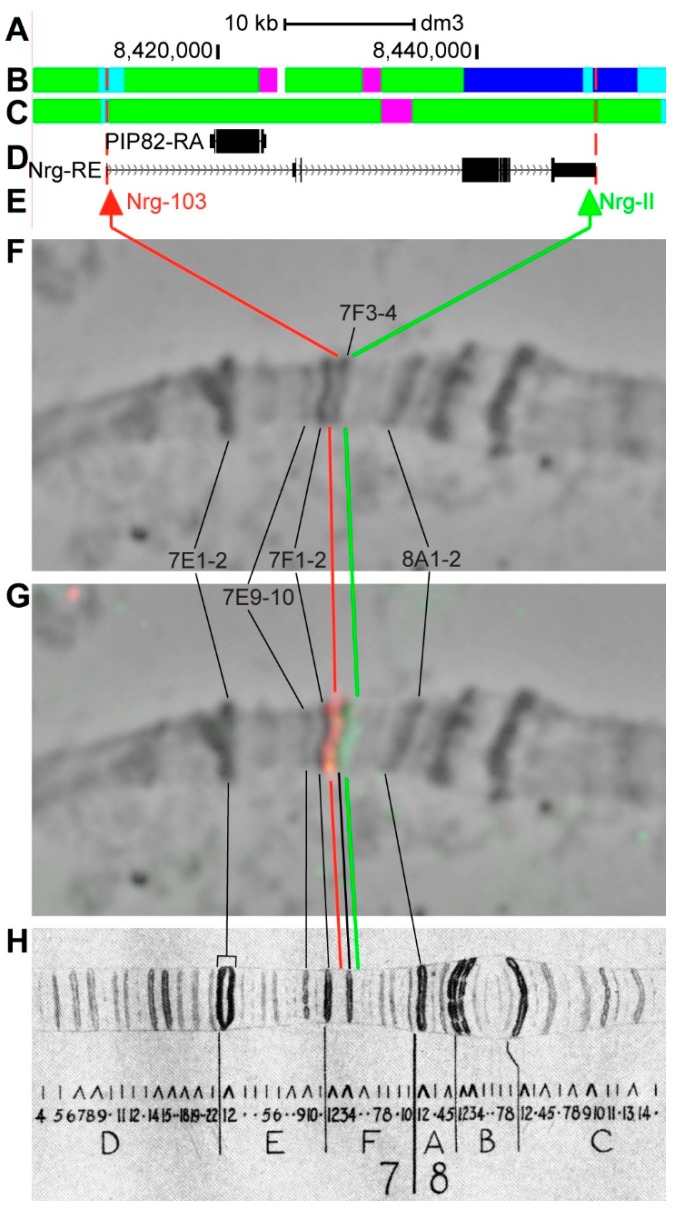
Localization of the *Nrg* gene in the 7F region (the gene is bound by red dotted lines). (**A**) the scale (kb) and genomic coordinates (bp); (**B**) the four chromatin-state model [39]; (**C**) the chromatin condensation level model [32]; (**D**) genes location (arrow lines show the gene orientation and black bars indicate the location of gene exons); (**E**) probes location; (**F**) phase-contrast micrograph of the region; (**G**) combined FISH signals; and, (**H**) C. Bridges’ chromosome map [25].

**Figure 15 genes-11-00417-f015:**
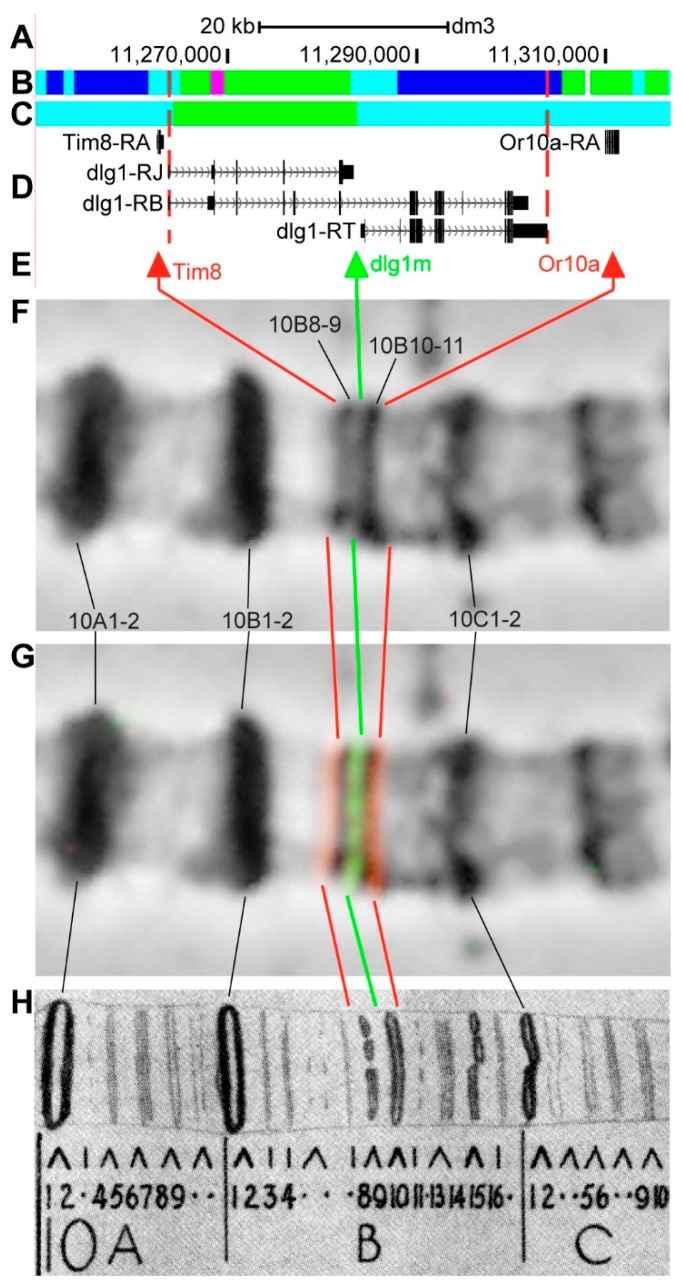
Localization of the *dlg1* gene in the 10B region (the gene is bound by red dotted lines). (**A**) the scale (kb) and genomic coordinates (bp); (**B**) the four chromatin-state model [39]; (**C**) the chromatin condensation level model [32]; (**D**) genes location (arrow lines show the gene orientation and black bars indicate the location of gene exons); (**E**) probes location; (**F**) phase-contrast micrograph of the region; (**G**) combined FISH signals; and, (**H**) C. Bridges’ chromosome map [25].

**Figure 16 genes-11-00417-f016:**
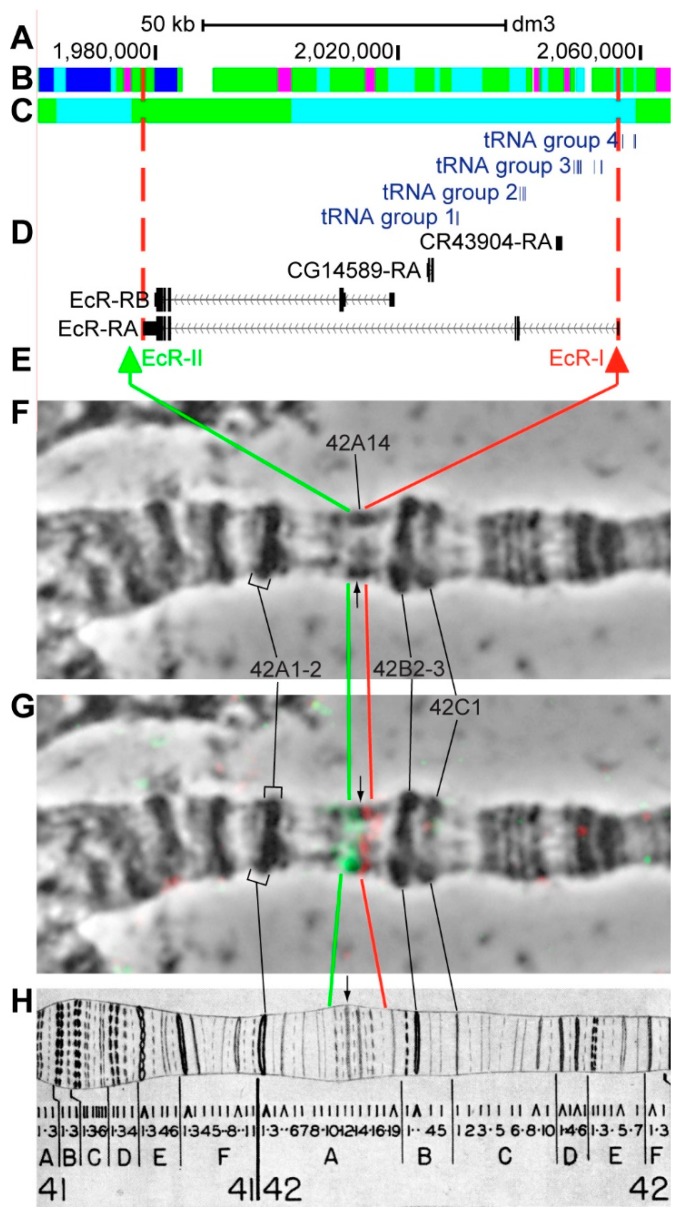
Localization of the *EcR* gene in the 42A region (the gene is bound by red dotted lines). (**A**) the scale (kb) and genomic coordinates (bp); (**B**) the four chromatin-state model [39]; (**C**) the chromatin condensation level model [32]; (**D**) genes location (arrow lines show the gene orientation and black bars indicate the location of gene exons); (**E**) probes location; (**F**) phase-contrast micrograph of the region; (**G**) combined FISH signals; and, (**H**) C. Bridges and P. Bridges’ chromosome map [55]. tRNA group 1 is represented by: *tRNA:R2:42Ad*, *tRNA:K2:42Ae*; tRNA group 2 is represented by: *tRNA:N5:42Ah*, *tRNA:N5:42Ag*, *tRNA:N5:42Af*; tRNA group 3 is represented by: *tRNA:R2:42Ac*, *tRNA:N5:42Ac*, *tRNA:N5:42Ab*, *tRNA:N5:42Aa*, *tRNA:K2:42Ac*, *tRNA:K2:42Ab*, *tRNA:K2:42Aa*; tRNA group 4 is represented by: *tRNA:CR30316*, *tRNA:I:42A*, *tRNA:R2:42Ab*, *tRNA:K2:42Ad*.

**Figure 17 genes-11-00417-f017:**
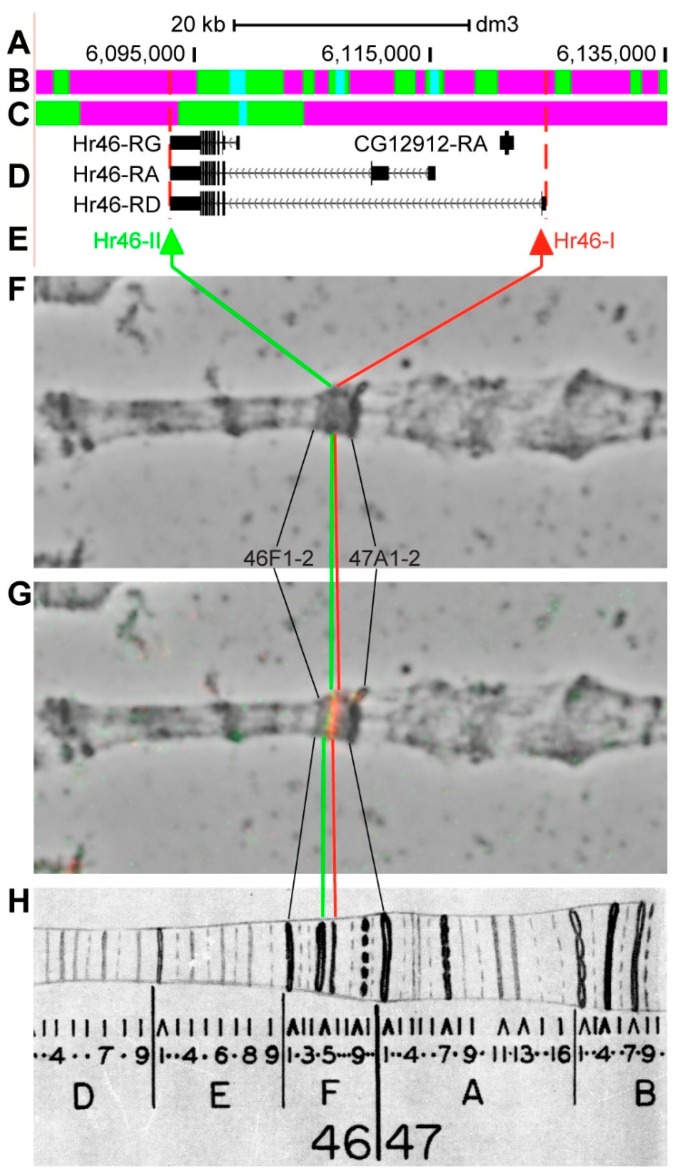
Localization of the *Hr46* gene in the 46F region (the gene is bound by red dotted lines). (**A**) the scale (kb) and genomic coordinates (bp); (**B**) the four chromatin-state model [39]; (**C**) the chromatin condensation level model [32]; (**D**) genes location (arrow lines show the gene orientation and black bars indicate the location of gene exons); (**E**) probes location; (**F**) phase-contrast micrograph of the region; (**G**) combined FISH signals; and, (**H**) C. Bridges’ chromosome map [25].

**Figure 18 genes-11-00417-f018:**
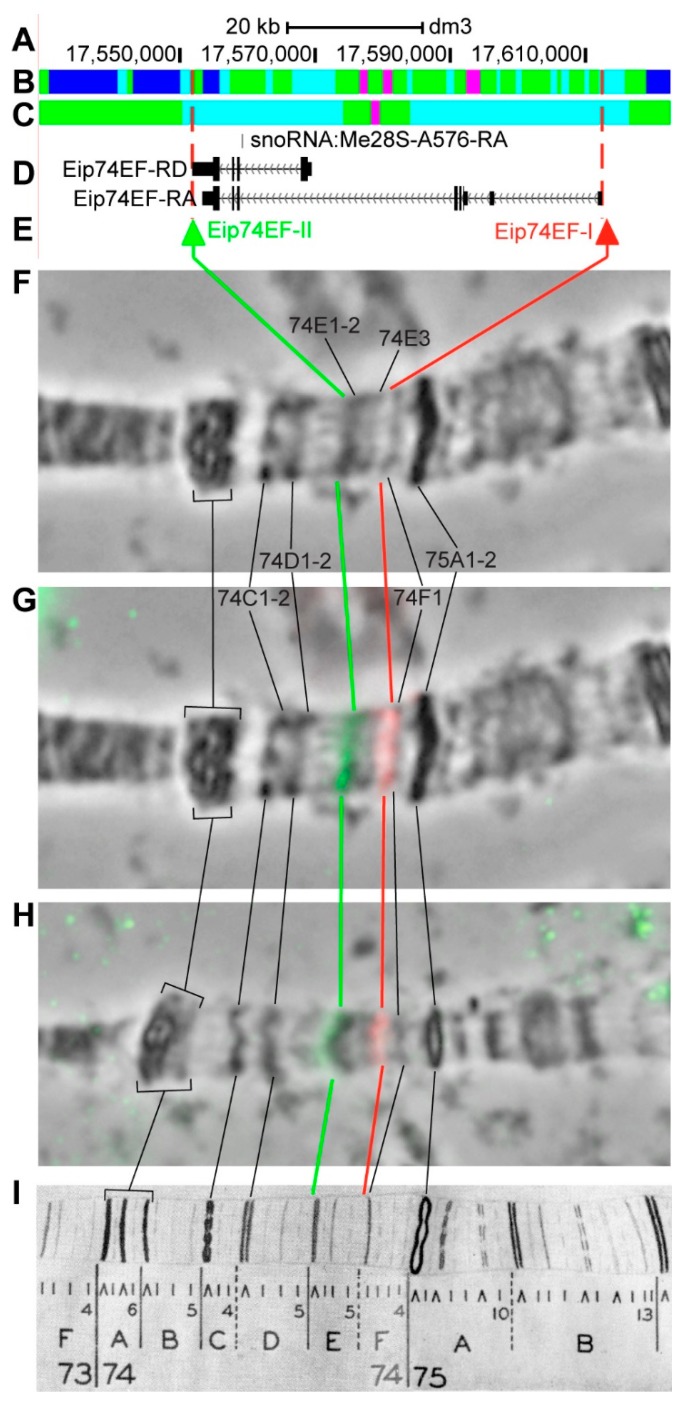
Localization of the *Eip74EF* gene in the 74E region (the gene is bound by red dotted lines). (**A**) the scale (kb) and genomic coordinates (bp); (**B**) the four chromatin-state model [39]; (**C**) the chromatin condensation level model [32]; (**D**) genes location (arrow lines show the gene orientation and black bars indicate the location of gene exons); (**E**) probes location; (**F**) phase-contrast micrograph of the region; (**G**) combined FISH signals; (**H**) combined FISH signals on the stretched chromosome preparation; and, (**I**) P. Bridges’ chromosome map [57].

**Figure 19 genes-11-00417-f019:**
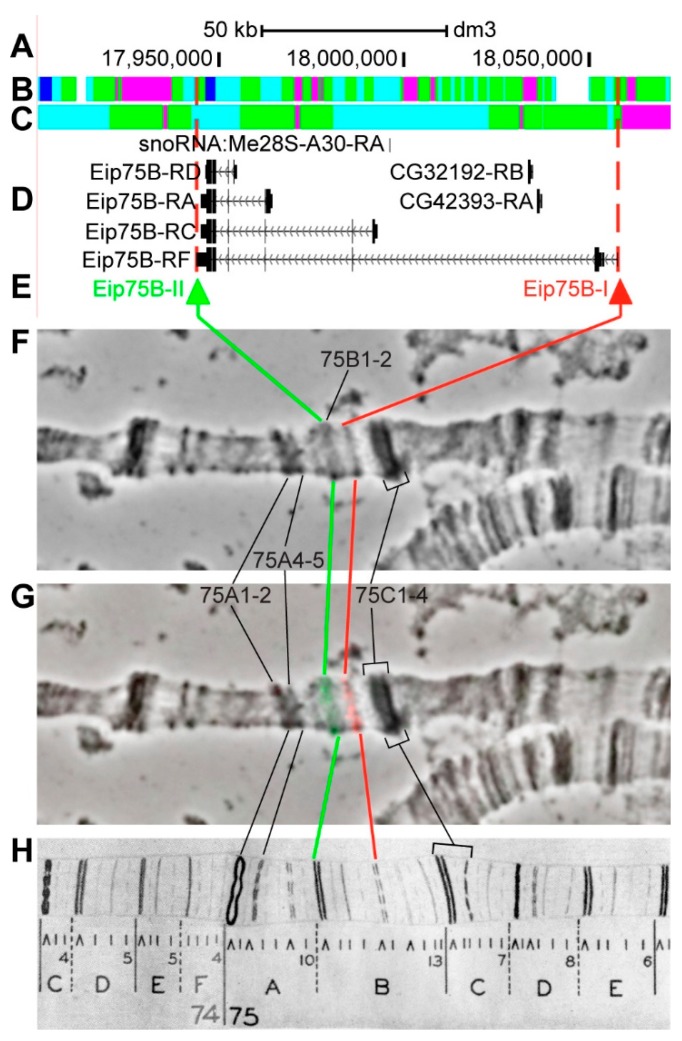
Localization of the *Eip75B* gene in the 75B region (the gene is bound by red dotted lines). (**A**) the scale (kb) and genomic coordinates (bp); (**B**) the four chromatin-state model [39]; (**C**) the chromatin condensation level model [32]; (**D**) genes location (arrow lines show the gene orientation and black bars indicate the location of gene exons); (**E**) probes location; (**F**) phase-contrast micrograph of the region; (**G**) combined FISH signals; and, (**H**) P. Bridges’ chromosome map [57].

**Figure 20 genes-11-00417-f020:**
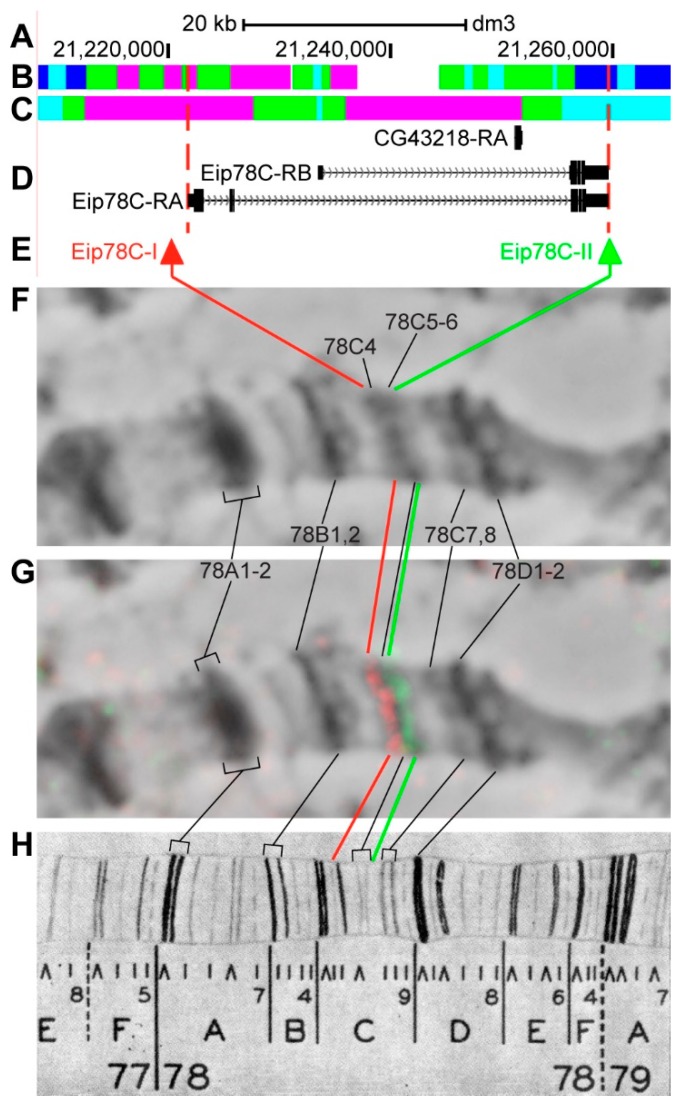
Localization of the *Eip78C* gene in the 78C region (the gene is bound by red dotted lines). (**A**) the scale (kb) and genomic coordinates (bp); (**B**) the four chromatin-state model [39]; (**C**) the chromatin condensation level model [32]; (**D**) genes location (arrow lines show the gene orientation and black bars indicate the location of gene exons); (**E**) probes location; (**F**) phase-contrast micrograph of the region; (**G**) combined FISH signals; and, (**H**) C. Bridges’ chromosome map [25].

**Figure 21 genes-11-00417-f021:**
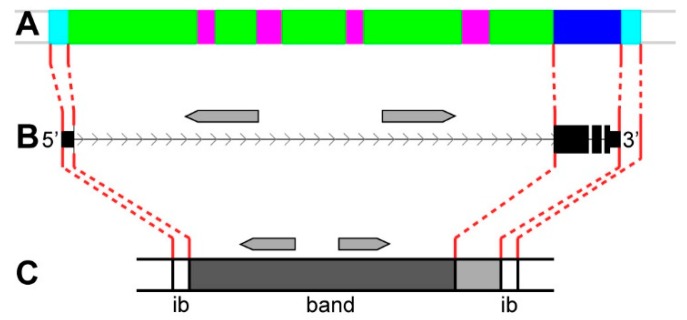
The scheme of the polytene chromosome band organization. The predominant long “main” gene contains exons and intron material; grey arrows represent shorter genes located within the introns of the “main” gene. (**A**) the four chromatin-state model [39]; (**B**) studied genes; and, (**C**) bands and interbands (ib) of polytene chromosome structure comprising the genes studied.

**Figure 22 genes-11-00417-f022:**
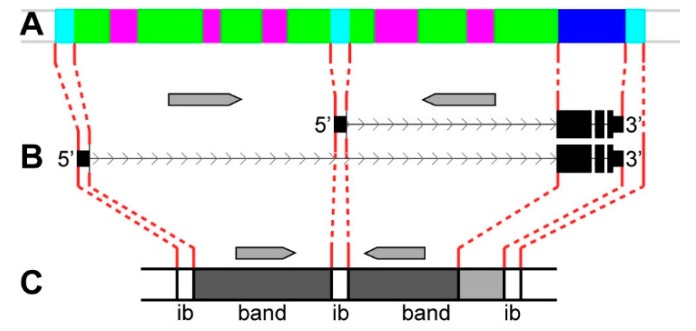
The scheme of the two bands and three interbands formation from the gene material with two introns and two alternative promoters of the predominant long “main” gene; grey arrows represent shorter genes located within the introns of the “main” gene. (**A**) the four chromatin-state model [39]; (**B**) exons and introns of the *dlg1* gene; and, (**C**) polytene chromosome bands and interbands (ib) structure comprising genes studied.

**Figure 23 genes-11-00417-f023:**
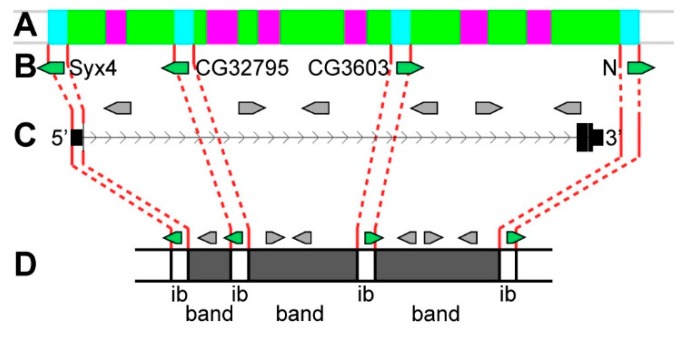
The scheme of the four bands and four interbands formation from the gene material with alternative promoters and three introns composed of late replicated chromatin; grey arrows represent shorter genes located within the introns of the predominant long “main” gene. (**A**) the four chromatin-state model [39]; (**B**) moderate level housekeeping genes *Syx4*, *CG32795*, *CG3603*, *N*; (**C**) exons and introns of the *kirre* gene; and, (**D**) polytene chromosome bands and interbands (ib) structure comprising “main”, short development, and housekeeping genes.

**Table 1 genes-11-00417-t001:** Fifteen long genes characteristics.

Gene	Gene Location According to UCSC	Gene Size, kb	Intron Size, %
*CG3777*	1A1–1A5	70.6	95.2
*CG43867*	1C5–1D2	119.7	94.2
*br (broad)*	2B3–2B4	70	93.5
*CG42666 (or prage)*	2B9–2B12	79.2	96.3
*trol (terribly reduced optic lobes)*	3A3–3A4	74.9	81.9
*sgg (shaggy)*	3A8–3B1	43.8	93.3
*kirre (kin of irre)*	3B4–3C7	393.7	98.4
*dnc (dunce)*	3C9–3D1	167.3	94.8
*Nrg (Neuroglian)*	7F2–7F4	37.7	80.4
*dlg1 (discs large 1)*	10B6–10B10	40.1	81.9
*EcR (Ecdysone receptor)*	42A9–42A12	78.6	93.5
*Hr46 (Hormone receptor-like in 46 or Hormone receptor 3)*	46F5–46F7	31.8	86.7
*Eip74EF (Ecdysone-induced protein 74EF)*	74D4–74E2	59.1	89.8
*Eip75B (Ecdysone-induced protein 75B)*	75A10–75B6	113.7	94.8
*Eip78C (Ecdysone-induced protein 78C)*	78C2–78C3	39.5	86.8

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
