# Peer review of "Genes Containing Long Introns Occupy Series of Bands and Interbands in Drosophila melanogaster Polytene Chromosomes"

_genes, 2020, doi:10.3390/genes11040417_

Round 1
Reviewer 1 Report
Khoroshko et al., present here a comprehensive mapping of long intron-containing genes onto Drosophila polytene chromosomes.
The study is well described and of interest for researchers in the field of genome organisation and regulation.
I have only minor comments to add:
The authors rely on the polytene chromosome maps/drawings of Bridges to identify the bands that are occupied by genes, since their phase-contrast micrographs are not of sufficient resolution. Could the authors explain how Bridges was able to achieve this high resolution in the mapping of bands but the authors could not? The authors should explain the method of microscoscopy and objectives (magnification, NA) and camera used for the phase-contrast micrographs. The authors compare the distribution of housekeeping genes as compared to developmentally regulated genes. What criteria are the authors using to define housekeeping genes in this study?Author Response
“Khoroshko et al., present here a comprehensive mapping of long intron-containing genes onto Drosophila polytene chromosomes.
The study is well described and of interest for researchers in the field of genome organization and regulation.
I have only minor comments to add:
The authors rely on the polytene chromosome maps/drawings of Bridges to identify the bands that are occupied by genes, since their phase-contrast micrographs are not of sufficient resolution. Could the authors explain how Bridges was able to achieve this high resolution in the mapping of bands but the authors could not?”
We use the Bridges’ maps of polytene chromosomes because these maps are standard and to this day no research group in the world could offer anything better. As for resolution in our microscopic figures, we show chromosomes captions after FISH procedures. As a rule, the resolution of the chromosome banding pattern after FISH falls down for thin bands critically, only thick bands stay clearly visible. Therefore, to show location of the thin bands we demonstrate the same chromosome regions on the Bridges’ map.
“The authors should explain the method of microscoscopy and objectives (magnification, NA) and camera used for the phase-contrast micrographs.”
Over the last few years we have been using digital microscope Olympus with X100 magnification of objective. We included the following information into the Materials and Methods section:
“Chromosome squashes were analyzed using epifluorescence optics (Olympus BX50 microscope) and photographed with CCD Olympus DP50 (objective X100). For every probe colocalization at least 50 nuclei on several slides were analyzed.”
“The authors compare the distribution of housekeeping genes as compared to developmentally regulated genes. What criteria are the authors using to define housekeeping genes in this study?”
The classification of the housekeeping genes have been previously discussed in other papers by our group. As information on it in the present study seems to be mentioned quite shortly, we added several corrections and updated the Introduction section.
We would like to express our gratitude to the Reviewer, the study has benefitted from their comments, suggestions and advice.

Reviewer 2 Report
In this manuscript Zhimulev and colleagues continue their longstanding efforts to understand the relationship between polytene chromosome structure (bands and interbands), and the underlying genome. This is a very classical but important endeavor considering that polytene chromosome morphology provides the most detailed direct picture of higher order chromosome structure available. Zhimulev is one of few investigators who has the background and cytogenetic skills to pursue this task at its highest level. Consequently, it is disappointing that the manuscript is flawed and could be much better.
Despite the nice but largely irrelevant historical introduction, the goals of the project are not clearly stated. Early researchers on this topic assumed that the pattern of bands and inter-bands seen in polytene chromosomes would correspond in some way to a linear sequence of genes with their exons, introns and UTRs that were presumed to make up chromosomes. The authors do not address what the relevant hypotheses are now that we know chromosomes contain multiple overlapping and nested genes, many with multiple promoters and transcripts, and surrounded by regulatory elements that may before, within or after the transcribed sequences. There is no mention of the importance of gene activity and what cell type is being studied. A major problem for this field is that most readers no longer see the relevance of the questions being asked.
The manuscript does not distinguish what is actually known and agreed upon by researchers in this field, from the personal beliefs of the Zhimulev lab. There is little discussion or citation of researchers who might have differing view. Since the modENCODE project researchers have divided chromatin into various classes based on localized features. The paper assumes that the particular chromatin types favored by this group are known facts, when few if any other researchers use this system and most continue to use variants of the ENCODE system. The authors also assume that the relationship of lab-favored chromatin types to polytene features has been proven- "aquamarine chromatin is found in interbands and contains promoters of housekeeping genes." But these types of conclusions are not actually known or proven. It remains very difficult to even define interbands experimentally.
The major conclusion of the paper seems to be that there are at least a handful of large genes that span a small number of bands/interbands. The in situ hybridization data, excellent though they are, are not convincing of that interpretation for some of the smaller examples, because it is not clear how much in situ signals span an extended region on the chromosome due to spreading of sequence copies during the process of squashing and stretching. In the largest genes, such as kirre, it is convincing that more the one band is included between the two physical ends of the longest kirre transcripts. However, there are 27 gene located internally, and the reason the gene is so long primarily is due to a distal promoter located within the syntaxin 4 gene that immediately splices back across much of the region. Is it enough to have a distant promoter, whose use is not discussed, to make all the internal region a gene? Or is kirre a oddball gene located in a single band with an aberrant transcript that starts from another gene? What do any of these findings tell us about the meaning of polytene chromosome banding structure?
Author Response
“In this manuscript Zhimulev and colleagues continue their longstanding efforts to understand the relationship between polytene chromosome structure (bands and interbands), and the underlying genome. This is a very classical but important endeavor considering that polytene chromosome morphology provides the most detailed direct picture of higher order chromosome structure available. Zhimulev is one of few investigators who has the background and cytogenetic skills to pursue this task at its highest level. Consequently, it is disappointing that the manuscript is flawed and could be much better.
Despite the nice but largely irrelevant historical introduction, the goals of the project are not clearly stated. Early researchers on this topic assumed that the pattern of bands and inter-bands seen in polytene chromosomes would correspond in some way to a linear sequence of genes with their exons, introns and UTRs that were presumed to make up chromosomes.”
For the historical part of the Introduction section, we would like to explain that our goal was to show that no researcher during 85 years after rediscovery of polytene chromosomes could have suggested any theoretical model according to which genes are situated in chromosomes the way we described in the present study. Never has the localization of long extended introns been taken into consideration when trying to elucidate the principle of the existing banding pattern organization.
“…the goals of the project are not clearly stated”
To address this comment, we added the following para to the Introduction section:
“As it was already mentioned, the length of an average band in Drosophila polytene chromosomes is about 30 kb according to various estimates [16,43], and the smallest distinguishable band contains 5 kb of DNA [44]. Compared with the DNA length in the bands, introns up to 400 kb are quite large and therefore must somehow be detected at the chromosome level; at least it can be expected that structures based on such long introns can be seen under the light microscope. According to our recently obtained preliminary data, certain large genes can occupy extended sections of chromosomes [45,46]. In the present study, we examined the localization sites of the genes mentioned above; in addition to that, we picked 12 new regions and analyzed their gene, intron, and chromatin composition, as well as studied their cytological location. We applied a simple approach, by using FISH we mapped the start and the end of each long gene (usually the longest transcript). Then we studied the distribution of transcripts, introns, exons and chromatin states. The data obtained show that the long genes and long introns can occupy several chromosome structures – interbands and different types of bands.”
“The authors do not address what the relevant hypotheses are now that we know chromosomes contain multiple overlapping and nested genes, many with multiple promoters and transcripts, and surrounded by regulatory elements that may before, within or after the transcribed sequences. There is no mention of the importance of gene activity and what cell type is being studied. A major problem for this field is that most readers no longer see the relevance of the questions being asked.”
In the recent studies we found no new ideas or explanation of evolutionary and functional gene/intron/band organization we have described in our study. Therefore, we limited the Discussion section to classification of the possible ways the genes and introns could be located in the bands.
“The manuscript does not distinguish what is actually known and agreed upon by researchers in this field, from the personal beliefs of the Zhimulev lab. There is little discussion or citation of researchers who might have differing view. Since the modENCODE project researchers have divided chromatin into various classes based on localized features. The paper assumes that the particular chromatin types favored by this group are known facts, when few if any other researchers use this system and most continue to use variants of the ENCODE system. The authors also assume that the relationship of lab-favored chromatin types to polytene features has been proven- "aquamarine chromatin is found in interbands and contains promoters of housekeeping genes." But these types of conclusions are not actually known or proven. It remains very difficult to even define interbands experimentally.”
During the last six years, we published several papers about distribution of genes in polytene chromosomes using our four-chromatin states model. It appeared as the result of experimental and mathematical treatment of the modENCODE data. All these questions are under discussion in some of the following papers:
Zhimulev IF, et al. 2014. PloS ONE 9(7):e101631
Khoroshko VA, et al. 2016. PloS ONE 11(6): e0157147.
Boldyreva LV, et al. 2017. Curr Genomics 18 (2): 214-226
Pokholkova GV, et al. 2018. PlosOne 13 (4): e0192634
Kolesnikova TD, et al. 2018. PLOS ONE 13 (4): e0195207
Zykova TYu, et al. 2018. Curr Genomics. 19 (3): 179-191
Sidorenko DS, et al. 2019. Chromosoma 128 (2): 97–117, 2019
Demakova OV, et al. 2019. Chromosoma, 2019, https://doi.org/10.1007/s00412-019-00728-2
Since the major portion of the references, assumptions and discussions have been addressed to in the previous studies, we decided not to include the already known data as the present manuscript is quite massive. Virtually all of the Reviewer’s comments are discussed in the articles mentioned above. Nevertheless, we added 40 new lines of text and updated the Introduction section in order to describe other chromatin models and the way our model differs from them. We also illustrate why our model allows to simultaneously relate the chromatin states to genes, bands and interbands, and how to divide the genes into housekeeping and development groups.
“The major conclusion of the paper seems to be that there are at least a handful of large genes that span a small number of bands/interbands. The in situ hybridization data, excellent though they are, are not convincing of that interpretation for some of the smaller examples, because it is not clear how much in situ signals span an extended region on the chromosome due to spreading of sequence copies during the process of squashing and stretching. In the largest genes, such as kirre, it is convincing that more the one band is included between the two physical ends of the longest kirre transcripts. However, there are 27 gene located internally, and the reason the gene is so long primarily is due to a distal promoter located within the syntaxin 4 gene that immediately splices back across much of the region. Is it enough to have a distant promoter, whose use is not discussed, to make all the internal region a gene? Or is kirre a oddball gene located in a single band with an aberrant transcript that starts from another gene?
What do any of these findings tell us about the meaning of polytene chromosome banding structure?”
In previous studies we showed that throughout the whole genome interbands are the sites where 5’-ends of housekeeping genes are located. After Francis Crick’ hypothesis we showed that the housekeeping genes occupy two chromosome structures – interbands and the neighboring grey bands. Then we studied the location of developmental genes in the black bands. And finally in the present study, for very long genes we show some new ways of cytological arrangement in the banding pattern of polytene chromosomes.
We would like to express our gratitude to the Reviewer, the study has benefitted from their comments, suggestions and advice.
Round 2
Reviewer 2 Report
The authors did not respond to the issues I raised in a satisfactory manner, but instead dismissed them via references to their own previous publications. However, without new and better answers to these issues than can be found in this or previous manuscripts, the significance of the reported work will be minimal.
Author Response
In our study we described the new principle of the polytene chromosome bands and interbands genetic organization. The Introduction and Discussion sections were written according to this finding in order to comprehensively cover the topic under investigation. Unfortunately the Reviewer 2 disregarded the importance of this new principle portrayal. Moreover he did not give any concrete criticism or indication of mistakes we could correct. Nevertheless, when addressing the issues that have been raised by the Reviewer 2 in Review Report (Round 1), we made a profound reconstruction of the article, which has benefitted greatly from their comments and suggestions. First of all, we added an extensive piece of information to the Introduction section (lines 98-127) and secondly, made minor proofs in different parts of the manuscript.
The following section was added to address the Reviewer’s comment that the study’s goal was not clearly stated (lines 134-145):
"As it was already mentioned, the length of an average band in Drosophila polytene chromosomes is about 30 kb according to various estimates [16,43], and the smallest distinguishable band contains 5 kb of DNA [44]. Compared with the DNA length in the bands, introns up to 400 kb are quite large and therefore must somehow be detected at the chromosome level; at least it can be expected that structures based on such long introns can be seen under the light microscope but such data are unavailable in literature. According to our recently obtained preliminary data, certain large genes can occupy extended sections of chromosomes [45,46]. In the present study, we examined the localization sites of the genes mentioned above; in addition to that, we picked 12 new regions and analyzed their gene, intron, and chromatin composition, as well as studied their cytological location. We applied a simple approach, by using FISH we mapped the start and the end of each long gene (usually the longest transcript). Then we studied the distribution of transcripts, introns, exons, chromatin states and location of other genes according to chromosome structures such as interbands, black and grey bands."
To conclude, we think that we answered all constructive and pertinent comments of the Reviewer 2, and would not agree with other points of his critiques. Moreover we cannot see which namely criticism we have to answer.